# Role of Nfu1 and Bol3 in iron-sulfur cluster transfer to mitochondrial clients

Andrew Melber[1,2], Un Na[1,2], Ajay Vashisht[3], Benjamin D Weiler[4], Roland Lill[4,5], James A Wohlschlegel[3], Dennis R Winge[1,2]*

[1]Department of Medicine, University of Utah Health Sciences Center, Salt Lake City, United States; [2]Department of Biochemistry, University of Utah Health Sciences Center, Salt Lake City, United States; [3]Department of Biological Chemistry, David Geffen School of Medicine at UCLA, Los Angeles, United States; [4]Institut für Zytobiologie, Philipps-Universität Marburg, Marburg, Germany; [5]LOEWE Zentrum für Synthetische Mikrobiologie SynMikro, Marburg, Germany

**Abstract** Iron-sulfur (Fe-S) clusters are essential for many cellular processes, ranging from aerobic respiration, metabolite biosynthesis, ribosome assembly and DNA repair. Mutations in *NFU1* and *BOLA3* have been linked to genetic diseases with defects in mitochondrial Fe-S centers. Through genetic studies in yeast, we demonstrate that Nfu1 functions in a late step of [4Fe-4S] cluster biogenesis that is of heightened importance during oxidative metabolism. Proteomic studies revealed Nfu1 physical interacts with components of the ISA [4Fe-4S] assembly complex and client proteins that need [4Fe-4S] clusters to function. Additional studies focused on the mitochondrial BolA proteins, Bol1 and Bol3 (yeast homolog to human BOLA3), revealing that Bol1 functions earlier in Fe-S biogenesis with the monothiol glutaredoxin, Grx5, and Bol3 functions late with Nfu1. Given these observations, we propose that Nfu1, assisted by Bol3, functions to facilitate Fe-S transfer from the biosynthetic apparatus to the client proteins preventing oxidative damage to [4Fe-4S] clusters.

*For correspondence: dennis. winge@hsc.utah.edu

**Competing interests:** The authors declare that no competing interests exist.

## Introduction

A severe syndrome characterized by the dysfunction of multiple mitochondrial enzymes has been described for a series of patients with mutations in four mitochondrial proteins IBA57, ISCA2, NFU1 and BOLA3 (*Seyda et al., 2001*; *Cameron et al., 2011*; *Navarro-Sastre et al., 2011*; *Ferrer-Cortès et al., 2013*; *Nizon et al., 2014*; *Baker et al., 2014*; *Debray et al., 2015*; *Lossos et al., 2015*; *Al-Hassnan et al., 2015*). Patients with this Multiple Mitochondria Dysfunctions Syndrome (MMDS) are afflicted with lactic acidosis, nonketotic hyperglycinemia and infantile encephalopathy typically leading to death in their first year of life. The syndrome is associated with an impairment of lipoic acid-dependent 2-oxoacid dehydrogenases arising from defective lipoate synthesis and defects in respiratory complexes I and II in select tissues including muscle and liver. These phenotypes arise from defective iron-sulfur (Fe-S) cluster assembly within the mitochondria. The deficiency in protein lipoylation is due to impaired activity of lipoic acid synthetase, which requires two [4Fe-4S] cluster cofactors (*Hiltunen et al., 2010*). The hyperglycinemic phenotype arises from failed lipoylation of the glycine cleavage enzyme. Whereas IBA57 and ISCA2 are known components of the ISA complex, along with ISCA1, which functions in the formation of [4Fe-4S] clusters within mitochondria (*Mühlenhoff et al., 2011*; *Gelling et al., 2008*; *Sheftel et al., 2012*), the functions of NFU1 and BOLA3 in Fe-S cluster assembly remain an enigma.

Yeast cells lacking Nfu1 are partially compromised in mitochondrial [4Fe-4S] cluster formation, but the defect is not as pronounced as in cells lacking components of the ISA complex (Isa1, Isa2 and

**eLife digest** Proteins perform almost all of the tasks necessary for cells to survive. Some of these proteins need to contain collections of iron and sulfur ions known as iron-sulfur clusters to work properly. The iron-sulfur clusters are first assembled from individual ions and then attached to the correct target proteins. In humans, yeast and other eukaryotic cells, the first step of this process happens in compartments called mitochondria and makes a cluster that contains two of each ion, known as [2Fe-2S] clusters. These [2Fe-2S] clusters can either be directly incorporated into target proteins, or they may be used to make larger iron-sulfur clusters – such as [4Fe-4S] clusters – in the mitochondria or the main compartment of the cell (the cytoplasm).

Defects that affect the assembly of proteins with iron-sulfur clusters are associated with severe diseases that affect metabolism, the nervous system and the blood. Mitochondria contain at least 17 proteins involved in making iron-sulfur proteins, but there may be others that have not yet been identified. For example, a study on patients with a rare human genetic disease suggested that proteins called BOLA3 and NFU1 might also play a role in this process.

Melber et al. used genetics to study how [4Fe-4S] clusters are assembled in the mitochondria of yeast cells. The experiments show that the yeast equivalents of NFU1 and BOLA3 (known as Nfu1 and Bol3) act to incorporate completed [4Fe-4s] clusters into their target proteins. This process is particularly important when iron-sulfur clusters are in high demand, such as when a cell needs to produce a lot of energy. Melber et al. also showed that a protein called Bol1 – which is closely related to Bol3 – is needed in an earlier stage of iron-sulfur cluster assembly.

The next steps following on from this work will be to look more closely at how Nfu1 and Bol3 deliver iron-sulfur clusters to the right target proteins. A future challenge will be to find out how other types of iron-sulfur clusters are transferred to their target proteins.

Iba57) (*Navarro-Sastre et al., 2011*; *Schilke et al., 1999*). As in patient cells with mutations in NFU1, yeast *nfu1Δ* cells have diminished protein lipoylation levels (*Navarro-Sastre et al., 2011*). Humans and yeast have two mitochondrial BolA proteins termed BolA1 (Bol1 in yeast) and BolA3 (Bol3 in yeast) (*Cameron et al., 2011*; *Willems et al., 2013*), but little is known concerning their physiological function. The similarities of phenotypes in patients with MMDS mutations in NFU1 and BOLA3 suggest that BOLA3 may likewise function in mitochondrial Fe-S biogenesis (*Cameron et al., 2011*).

Fe-S cluster synthesis within the mitochondria occurs on a scaffold complex and preformed clusters are subsequently transferred to recipient proteins (*Lill et al., 2012*). The initial cluster formed is a [2Fe-2S] cluster assembled on the ISU scaffold complex consisting of five proteins, Nfs1, Isd11, Yfh1, Yah1 and Isu1 (or Isu2; yeast nomenclature) (*Lill et al., 2012*; *Schmucker et al., 2011*; *Tsai and Barondeau, 2010*; *Lange et al., 2000*; *Webert et al., 2014*). The sulfide ions are provided by the Nfs1 cysteine desulfurase, along with its effector proteins Isd11 and Yfh1 (*Tsai and Barondeau, 2010*; *Lill and Mühlenhoff, 2008*; *Gerber et al., 2003*; *Biederbick et al., 2006*; *Bridwell-Rabb et al., 2014*; *Parent et al., 2015*; *Fox et al., 2015*). Assembled [2Fe-2S] clusters on Isu1 are transferred to the monothiol glutaredoxin Grx5 through the action of the Ssq1 ATPase and the DnaJ protein Jac1 (*Ciesielski et al., 2012*; *Majewska et al., 2013*; *Uzarska et al., 2013*). Two [2Fe-2S] clusters transferred by Grx5 are condensed into a [4Fe-4S] cluster on the downstream ISA complex (Isa1, Isa2 and Iba57) prior to transfer to client proteins (*Mühlenhoff et al., 2011*; *Gelling et al., 2008*; *Sheftel et al., 2012*; *Brancaccio et al., 2014*).

Nfu1 has been implicated to function as a late Fe-S maturation factor in bacteria and fungi (*Navarro-Sastre et al., 2011*; *Bandyopadhyay et al., 2008*; *Py et al., 2012*), an alternate scaffold protein for cluster synthesis (*Cameron et al., 2011*; *Tong et al., 2003*) or as a persulfide reductase associated with the sulfide transfer (*Liu et al., 2009*). The lack of NfuA in *Escherichia coli* and *Azotobacter vinelandii* is associated with decreased viability under stress conditions (*Bandyopadhyay et al., 2008*; *Py et al., 2012*; *Angelini et al., 2008*). Nfu proteins from most species are multidomain proteins. *E. coli* NfuA and human Nfu1 are two domain proteins with the C-terminal domain containing the functionally important CxxC motif that is known to bind a [4Fe-4S] cluster at a homodimer interface (*Bandyopadhyay et al., 2008*; *Tong et al., 2003*; *Angelini et al.,*

2008; *Gao et al., 2013*). The N-terminal domains differ between the *E. coli* and human proteins and lack a related CxxC motif. Recombinant expression and purification of *Azotobacter* NfuA or human Nfu1 did not result in Fe-S cluster bound to the purified protein, but in vitro Fe-S reconstitution studies followed by Mössbauer spectral studies demonstrated the presence of a [4Fe-4S] cluster (*Bandyopadhyay et al., 2008*; *Py et al., 2012*; *Tong et al., 2003*; *Angelini et al., 2008*). *Synechocystis* NifU was reported to bind a [2Fe-2S] cluster (*Yabe et al., 2004*; *Nishio and Nakai, 2000*), but Mössbauer spectral studies were not done to validate the assignment. The ability of Nfu1 to bind a [4Fe-4S] cluster supported the suggestions that Nfu1 was either an alternative scaffold protein involved in Fe-S cluster formation or involved in a late cluster transfer step. The ability of bacterial NfuA to transfer its cluster to apo-aconitase in vitro is consistent with a role in a late step of cluster transfer (*Bandyopadhyay et al., 2008*; *Angelini et al., 2008*).

BolA proteins are also known to coordinate Fe-S clusters in conjunction with monothiol glutaredoxins (*Li and Outten, 2012*). One of the three BolA proteins in *Arabidopsis thaliana* BolA1 was shown to bind a [2Fe-2S] cluster in a complex with glutaredoxin (Grx) (*Roret et al., 2014*). The cluster associated with the BolA:Grx complex is coordinated by two thiolate ligands, one from Grx and the other from an associated glutathione, and two histidine ligands from BolA1. Likewise, the cytosolic BolA2 proteins of yeast and humans coordinate [2Fe-2S] clusters at the heterodimer interface with monothiol glutaredoxins (*Li and Outten, 2012*; *Li et al., 2012*). Little is known about the physiological function of mitochondrial BolA proteins, designated Bol1 and Bol3. BolA proteins are found only in aerobic species (*Willems et al., 2013*). Depletion of the mitochondrial BolA1 in HeLa cells caused an oxidative shift in the mitochondrial thiol/disulfide redox ratio (*Willems et al., 2013*).

We set out to define the functional steps of Nfu1 and two mitochondrial BolA proteins in yeast. We report that Nfu1 and Bol3 function at a late step in the transfer of Fe-S clusters from the ISA complex to mitochondrial client proteins as a protective measure for [4Fe-4S] clusters from oxidative stress damage. In contrast to Bol3, the related mitochondrial Bol1 shows an interaction with Grx5 but not with the ISA complex or [4Fe-4S] client proteins.

## Results

### Nfu1 is associated with mitochondrial [4Fe-4S] cluster formation

*S. cerevisiae* cells lacking the mitochondrial Nfu1 protein (*nfu1Δ* cells) are markedly impaired in growth on synthetic complete medium with acetate as a carbon source (*Figure 1A*). However, the mutant cells display only a slight growth impairment on glycerol/lactate medium, suggesting a partial respiratory growth defect that is exacerbated with acetate as the carbon source. It was previously reported that *nfu1Δ* cells exhibit specific but partial defects in the formation of [4Fe-4S] clusters analogous to phenotypes seen in patients with mitochondrial dysfunction syndrome (*Navarro-Sastre et al., 2011*; *Schilke et al., 1999*). We confirmed the defects in [4Fe-4S] client enzymes reported for *nfu1Δ* cells showing that aconitase and succinate dehydrogenase (SDH) activities are markedly impaired, yet residual activity persists (*Figure 1B*). Aconitase activity is markedly attenuated in *nfu1Δ* yeast cells, whereas its activity is not significantly depleted in human *nfu1* patients (*Cameron et al., 2011*; *Navarro-Sastre et al., 2011*). No defect was observed in the yeast mutant in respiratory complex III, cytochrome $bc_1$, which requires a [2Fe-2S] cluster in its Rieske Rip1 subunit or in cytochrome oxidase that requires a [2Fe-2S] cluster in Yah1 for heme *a* formation (*Figure 1B*).

Consistent with the known defects of *nfu1Δ* yeast cells and human *nfu1* patients, lipoic acid (LA) conjugates on pyruvate dehydrogenase (PDH) and oxoglutarate dehydrogenase (KDH) were attenuated in *nfu1Δ* cells (*Figure 1C*) (*Navarro-Sastre et al., 2011*). As mentioned, lipoic acid formation is dependent on the [4Fe-4S] lipoic acid synthase Lip5 (*Hiltunen et al., 2010*). Steady-state protein analysis by SDS-PAGE showed diminished Sdh2 levels, the Fe-S subunit of SDH. Sdh2 contains three distinct Fe-S clusters ([2Fe-2S], [4Fe-4S], and [3Fe-4S] clusters), which transfer electrons from the catalytic Sdh1 subunit to ubiquinone. In the absence of Fe-S cluster insertion, Sdh2 stability is compromised (*Kim et al., 2012*) (*Figure 1C*) In contrast, the aconitase protein stability is not dependent on the presence of its [4Fe-4S] cluster (*Gelling et al., 2008*).

Two enzymes involved in yeast lysine biosynthesis Aco2 and Lys4 contain [4Fe-4S] clusters (*Fazius et al., 2012*). Whereas yeast lacking the ISA complex are auxotrophic for lysine and accumulate homocitrate as a metabolic intermediate, *nfu1Δ* cells propagate normally in medium lacking lysine

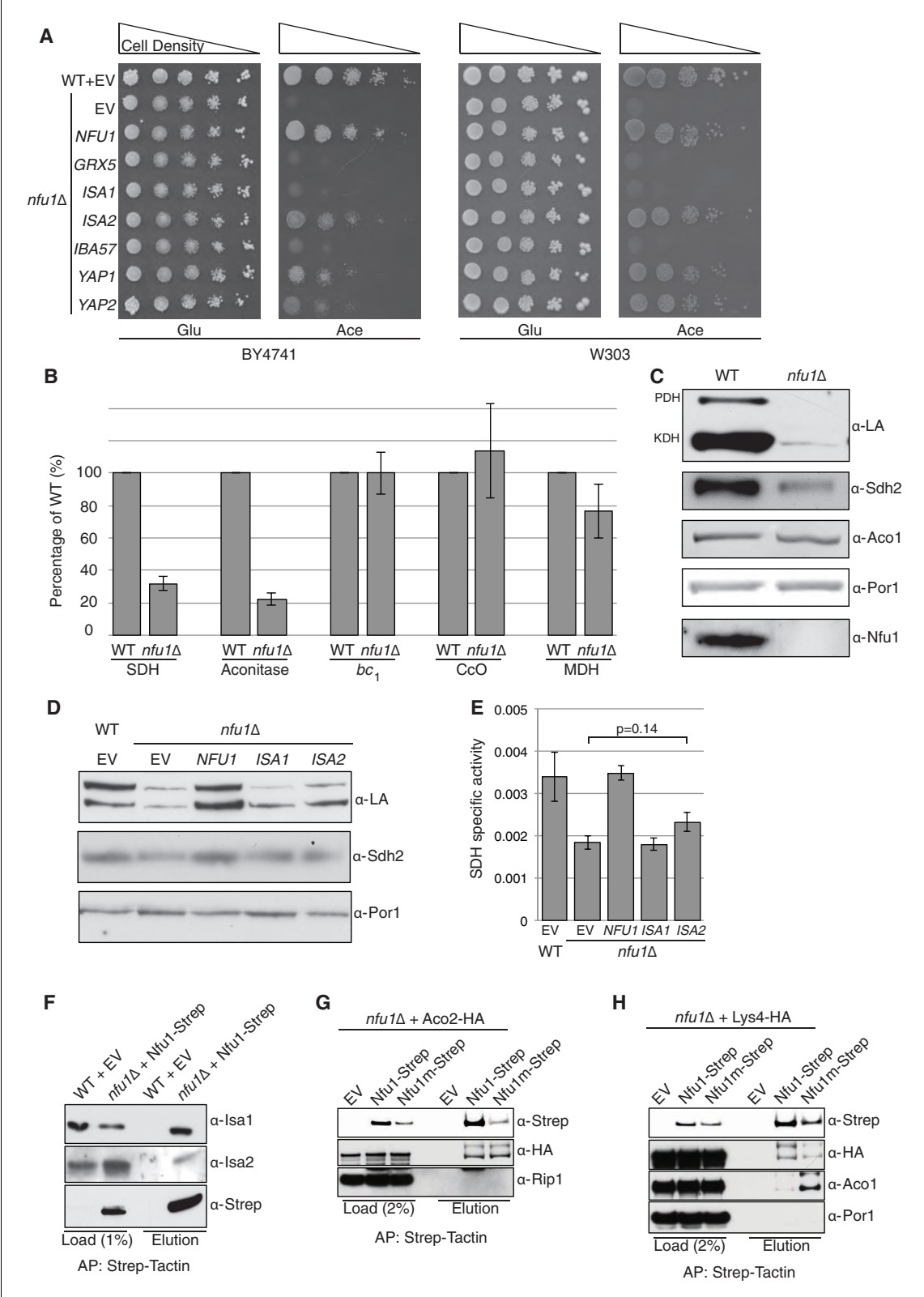

**Figure 1.** Nfu1 functions with both the ISA [4Fe-4S] assembly complex and [4Fe-4S] client proteins. Cells lacking Nfu1 exhibit defects in [4Fe-4S] cluster enzymes in mitochondria. (A) Respiratory growth defects revealed by yeast drop-test. Cells harboring empty vectors (EV) or high-copy plasmids expressing designated genes were pre-cultured in liquid synthetic complete (SC) glucose media lacking uracil. Serially diluted cells (10-fold) were spotted on SC media plates at 30°C. Grx5 is a monothiol glutaredoxin involved in mitochondrial Fe-S biogenesis. Isa1, Isa2 and Iba57 are subunits of

*Figure 1 continued on next page*

Figure 1 continued

the ISA scaffold complex required for [4Fe-4S] cluster synthesis. Yap1 is a transcription factor that induces expression of anti-oxidant genes. Glu is 2% glucose and Ace is 2% acetate. (B) The relative activity of aconitase, SDH, cytochrome $bc_1$, cytochrome $c$ oxidase (CcO), and malate dehydrogenase (MDH) were measured in isolated mitochondria from cells cultured in SC media with 2% raffinose. Data are shown as mean ± SE (n = 3) (CcO, n = 4). (C) Steady-state protein levels measured by SDS-PAGE followed by immunoblotting in isolated mitochondria. Anti-LA antibody is an antibody specific to lipoic acid (LA) that is conjugated to proteins. PDH is pyruvate dehydrogenase and KDH is α-ketoglutarate dehydrogenase. Sdh2 is the Fe-S cluster subunit of SDH. Aco1 is mitochondrial aconitase. Por1 is a mitochondrial loading control. (D) Restoration of LA moieties on PDH and KDH shown by SDS-PAGE followed by immunoblotting in isolated mitochondria from *nfu1Δ* cells over-expressing *ISA1* and *ISA2*. (E) Enzymatic activity of SDH in mitochondria isolated from *nfu1Δ* cells over-expressing *ISA1* and *ISA2*. Data are shown as mean ± SE (n = 3). (F) Strep-tag affinity purification of Nfu1-Strep revealed the Nfu1 interaction with Isa1 and Isa2. Mitochondria were solubilized with 0.1% n-dodecyl maltoside (DDM). Clarified lysates were incubated with Strep-Tactin superflow beads for 16 hr. After washing, proteins were eluted with 2.5 mM desthiobiotin, and then analyzed by immunoblotting. (G) Strep-tag affinity purification of Nfu1-Strep in the presence of ectopically expressed Aco2-HA. Nfu1m-Strep is the G/T>H mutant described in *Figure 4*. (H) Strep-tag affinity purification of Nfu1-Strep in the presence of ectopically expressed Lys4-HA. Lys4 and Aco2 are both nuclear DNA-encoded mitochondrial proteins that require a [4Fe-4S] cluster for each function in the lysine biosynthetic pathway in yeast. Nfu1m-Strep is the G/T>H mutant described in *Figure 4*.

and do not accumulate homocitrate as shown by GC-MS metabolomic studies (data not shown). Thus, sufficient [4Fe-4S] cluster synthesis and distribution occurs in *nfu1Δ* cells for lysine synthesis.

The growth defect of *nfu1Δ* yeast cells on acetate medium was severe, creating an opportunity to conduct screening for genetic suppressors of the respiratory defect. In a screen using transformants with a high-copy yeast DNA library, we isolated respiratory competent vector-borne clones of *nfu1Δ* BY4741 cells containing *NFU1, ISA2*, and the *YAP2* transcriptional activator. Each gene was recloned into yeast vectors and *nfu1Δ* transformants of both BY4741 and W303 genetic backgrounds were analyzed for growth on acetate medium for respiratory function. Although Isa2 is a component of the mitochondrial ISA heterotrimeric complex comprised of Isa1, Isa2 and Iba57, overexpression of Isa2 was the only ISA component capable of partially restoring respiratory growth of *nfu1Δ* cells on acetate medium (*Figure 1A*). *ISA2* transformants of *nfu1Δ* cells showed a partial restoration lipoylation of KDH and SDH activity suggesting that the respiratory capacity of the mutant cells was partially restored by elevated Isa2 levels (p value ~0.14) (*Figure 1D and E*). Thus, the respiratory function of Nfu1 can be partially replaced by super-physiological levels of the Isa2 component of the ISA complex.

## Nfu1 binds the ISA complex and [4Fe-4S] client proteins

An association of Nfu1 with the mitochondrial ISA complex was suggested by the observed suppression of the respiratory defect of *nfu1Δ* cells by *ISA2* overexpression along with defects in [4Fe-4S] mitochondrial enzymes. We tested if Nfu1 physically interacts with the ISA complex by co-immunoprecipitation studies using a functional C-terminal Strep tagged chimera of Nfu1. Affinity purification of Nfu1-Strep with Strep-Tactin beads showed co-purification of Isa1 and Isa2 (*Figure 1F*). In addition to the interaction with Isa1 and Isa2, Nfu1 associated with three [4Fe-4S] client proteins Aco1, Aco2 and Lys4, but not the [2Fe-2S] client protein Rip1 (*Figure 1G and H*).

## Nfu1 is necessary for protecting Fe-S clusters from oxidative damage

The partial respiratory function of *nfu1Δ* cells was also restored by overexpression of Yap2 or its paralogue Yap1 (*Figure 1A*). Yap1 and Yap2 are transcriptional activators that induce the expression of a battery of antioxidant genes, including thioredoxin, thioredoxin reductase and glutathione reductase, in response to oxidative stress (*Fernandes et al., 1997*). To confirm that the suppression of *nfu1Δ* cells by the YAP transcription factors was specifically due to a recovery of the [4Fe-4S] centers, we analyzed mitochondria from the transformants to test for restoration of lipoic acid conjugates of PDH and KDH and observed a clear restoration of LA-associated PDH (*Figure 2A*). The identification of *YAP1* and *YAP2* as high copy suppressors of *nfu1Δ* cells suggested a role for Nfu1 during oxidative stress. Consistent with this postulate, the respiratory growth of *nfu1Δ* cells was partially restored with the addition of the antioxidants, GSH and N-acetyl cysteine (NAC) to the growth medium (*Figure 2B*). These results support a role for Nfu1 during oxidative metabolism.

Since Nfu1 is important under oxidative conditions, we tested whether Nfu1 is dispensable during anoxic growth. WT and *nfu1Δ* cells were cultured to mid-log growth in normoxic or anoxic

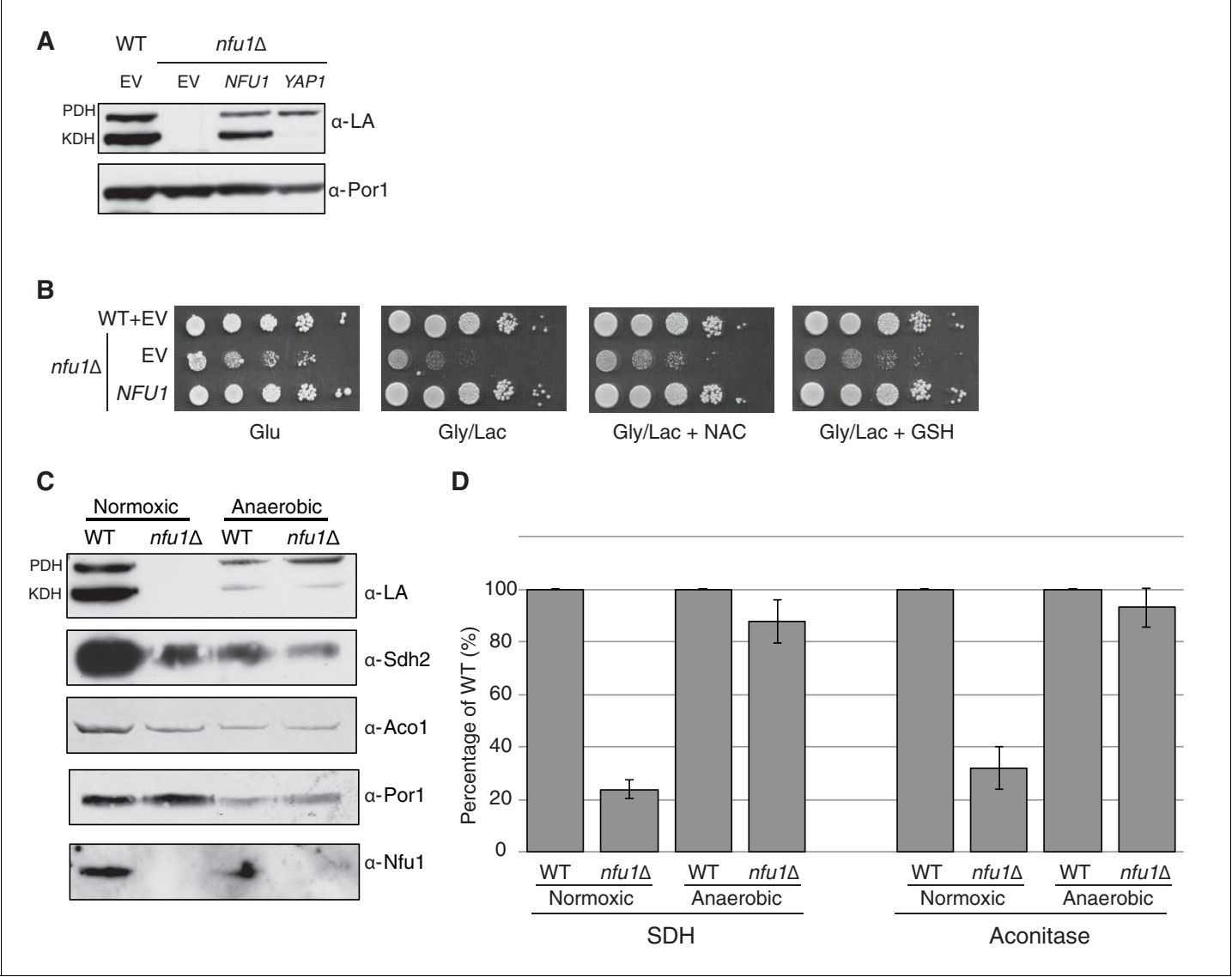

**Figure 2.** Nfu1 has a heighted importance during times of oxidative stress and is expendable in anoxic conditions. Defects in cells lacking Nfu1 are pronounced under oxidative stress conditions. (**A**) Steady-state levels of proteins in isolated mitochondria from *nfu1Δ* cells harboring high-copy *NFU1* plasmids or *YAP1* plasmids. (**B**) Yeast drop-test with 5 mM n-acetyl cysteine (NAC) and 2 mM glutathione (GSH). Gly/Lac is SC medium with 2% glycerol and 2% lactate as carbon sources. (**C**) Steady-state levels of proteins in isolated mitochondria from cells cultured under normoxic conditions or anaerobic conditions. (**D**) Relative activity of SDH and aconitase in mitochondria from panel **C**. Data are shown as mean ± SE (n = 3).

conditions. Mitochondria isolated from the cells were analyzed by steady-state protein analysis and enzymatic function of various [4Fe-4S] cluster enzymes. As previously described normoxic *nfu1Δ* cells exhibited the expected marked attenuation in SDH and aconitase activities and reduced lipoic acid adducts; however, the anoxic cells did not exhibit a significant difference between WT and *nfu1Δ* cells (*Figure 2C and D*). It should be noted that anoxic WT cells showed a marked reduction in mitochondrial enzymatic activities and steady-state protein levels compared to normoxic WT cells (~30% of normoxia), yet anoxic *nfu1Δ* cells did not show a marked further attenuation in SDH and lipoic acid conjugates. Thus, the cells are more dependent on Nfu1 during oxidative metabolism.

## The NfuC domain of Nfu1 harbors a CxxC motif required for function

Nfu1 consists of two domains in addition to the N-terminal mitochondrial targeting sequence (MTS) based on sequence homologies (*Figure 3A*). The N-terminal domain (NfuN, residues 22–126) is only

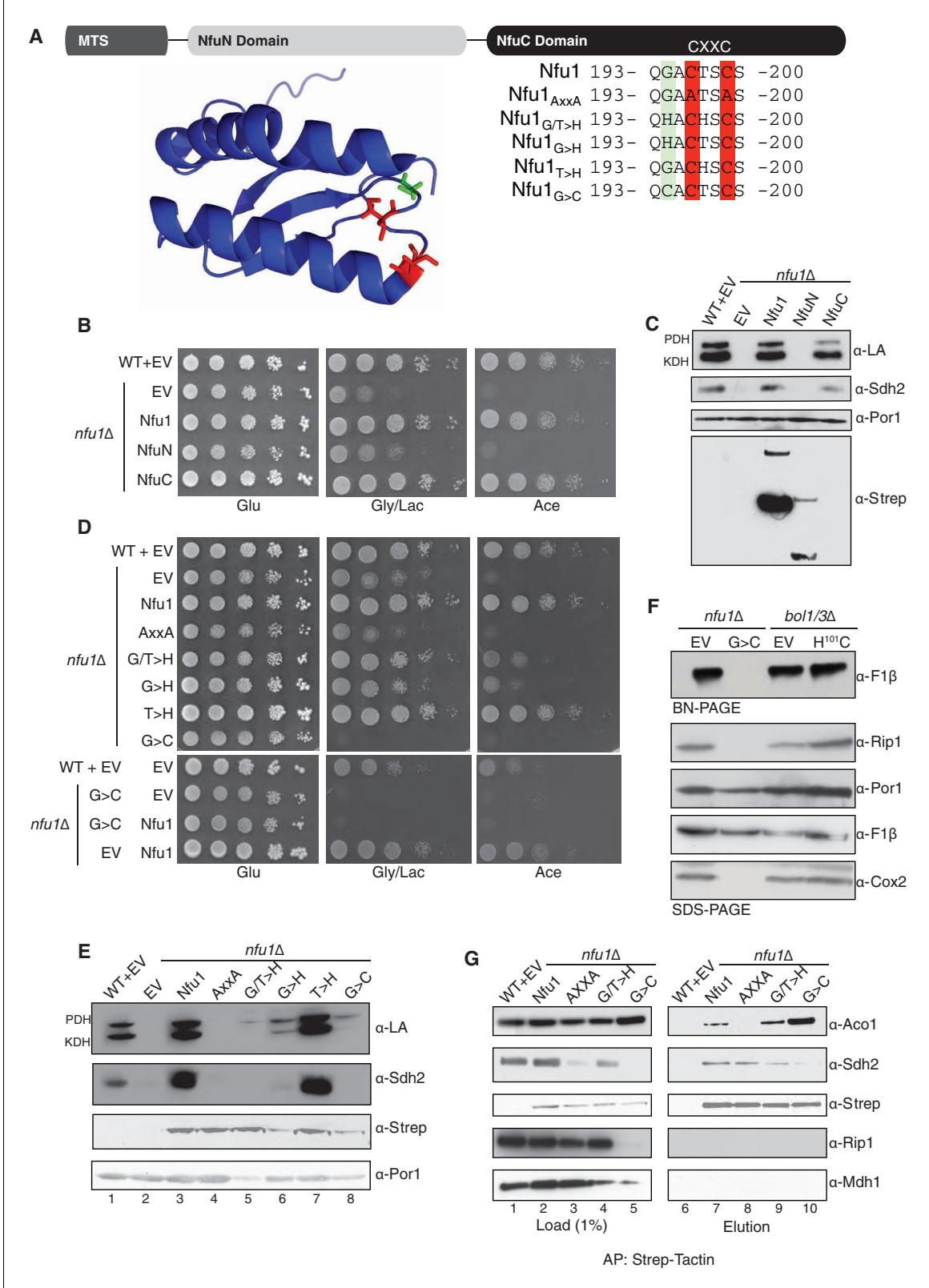

**Figure 3.** The CxxC motif of C-terminal domain of Nfu1 is essential for function. The CxxC motif is critical for Nfu1 function. (**A**) A schematic representation of Nfu1 domains. MTS, the mitochondrial targeting sequence; NfuN, the N-terminal domain of Nfu1; NfuC, the C-terminal domain harboring the highly conserved CxxC motif. The human NfuC tertiary structure (PDB: 2M5O) and primary sequences showing the CxxC motif (red) and adjacent amino acids indicated in partial sequences (green). (**B**) The respiratory growth defect of *nfu1Δ* cells was rescued with NfuC. Nfu1, NfuN, and

*Figure 3 continued on next page*

*Figure 3 continued*

NfuC were all fused with a C-terminal Strep-tag and expressed exogenously using low-copy plasmids. (C) Restoration of Nfu1 target proteins by NfuC expression in *nfu1Δ* cells. (D) Respiratory growths of *nfu1Δ* cells that express Nfu1 sequence variants were tested. All variants were fused with a Strep-tag and expressed on low-copy plasmids. (E) Steady-state levels of LA-conjugated proteins and Sdh2 in *nfu1Δ* cells that express Nfu1 variants. (F) BN-PAGE and SDS-PAGE analysis of [4Fe-4S] cluster independent enzymes in the dominant negative backgrounds *nfu1Δ* + G>C and *bol1/3Δ* + H$^{101}$C. (G) Strep-tag purification of Nfu1 sequence variants as described in *Figure 3B* immunoblotting for [4Fe-4S] cluster client proteins Aco1 and Sdh2.

The following figure supplements are available for figure 3:

**Figure supplement 1.** Yeast growth tests evaluating respiratory growth (Gly/Lac) of *nfu1Δ* + G>C cells following treatment with 5'-Fluoroorotic acid (5-FOA) to show cells have not lost their mitochondrial DNA (rho⁻).

**Figure supplement 2.** Affinity purification using Strep-Tactin to immobilize Strep tagged Nfu1 and the Nfu1 AxxA variant expressed ectopically in the BY4743 background with a single copy of Lys4 chromosomally tagged with GFP.

conserved within eukaryote species, while the C-terminal NifU-like domain (NfuC, residues 143–256) is widely conserved in all species and contains the important the Fe-S binding CxxC motif (*Figure 3A*). To test the functional importance of the two domains, both domains were separately expressed in *nfu1Δ* cells with the endogenous MTS of Nfu1 (1–21) to ensure proper delivery to the mitochondrial matrix.

Cells containing only the Nfu1 NfuC domain were capable of respiratory growth on either glycerol/lactate or acetate medium (*Figure 3B*), whereas cells harboring only the Nfu1N domain failed to propagate. Additionally, cells with the NfuC, but not the NfuN, domain showed normal Sdh2 and lipoic acid levels. Although the NfuN domain failed to restore Nfu1 function, the fragment was well expressed in cells, unlike the functional C-terminal domain that was markedly attenuated in protein stability (*Figure 3C*). The functionality of the NfuC domain suggests that only minimal levels of Nfu1 are important for function. The NfuN domain exhibited a putative dimeric species, analogous to the intact Nfu1 on the denaturing gels.

To further address the functional importance of the NfuC domain, we generated a series of amino acid substitutions within and near the conserved CxxC motif to the full-length Nfu1 protein (*Figure 3A*). One Nfu1 variant generated had the two cysteinyl residues in the CxxC motif (highlighted in red in *Figure 3A*) replaced with alanines. Cells harboring Nfu1 with the two CxxC cysteinyl residues replaced by alanines exhibited a respiratory growth defect analogous to *nfu1Δ* cells suggesting a loss-of-function phenotype (*Figure 3D and E*). The critical role of the CxxC motif cysteines was previously shown in the *E. coli* NfuA (*Angelini et al., 2008*).

A conserved glycine just upstream of the CxxC motif is commonly mutated to a cysteine in patients with MMDS (*Navarro-Sastre et al., 2011*; *Nizon et al., 2014*). We generated amino acid substitution of this Gly to Cys or His residues and replaced the conserved threonine between the two Cys residues by a His. Each mutant of Nfu1 was expressed in *nfu1Δ* cells and tested for function. The most striking substitution was the G>C mutant that mimics the MMDS1 patient allele, which displayed a severe synthetic sick phenotype on glycerol/lactate medium (*Figure 3D*). SDH biogenesis was markedly decreased and in addition Rip1 levels were low suggesting a block in *bc₁* biogenesis (*Figures 3E and G*, lane 5). This dominant negative phenotype was reversed when cells were plated on medium containing 5-fluoroorotic acid (5-FOA) to shed the *URA3*-containing plasmid harboring the G$^{194}$C Nfu1 mutant (*Figure 3—figure supplement 1*). Thus, the synthetic phenotype did not arise from mtDNA loss or any other irreversible pleiotropic defects. In addition, co-expression of a wild-type Nfu1 with the G$^{194}$C Nfu1 mutant failed to restore respiratory growth, demonstrating the dominant negative nature of this mutant (*Figure 3D*, bottom panel).

Although *nfu1Δ* cells with the G$^{194}$C mutant retained its mtDNA, mitochondrial translation was likely impaired due to attenuated levels of the assembled $F_1F_0$ ATPase on BN-PAGE and Cox2 steady-state levels (*Figure 3F*). Although the assembled $F_1F_0$ ATPase complex is markedly diminished, the steady-state levels of Atp2 in the $F_1$ sector are normal. Cells impaired in lipoic acid formation are deficient in tRNA processing by RNase P leading to attenuation in mitochondrial translation (*Schonauer et al., 2008*; *Hiltunen et al., 2009*). Diminished mitochondrial translation of the *bc₁* cytochrome *b* subunit would account for the reduced Rip1 levels observed (*Figure 3F and G*). In

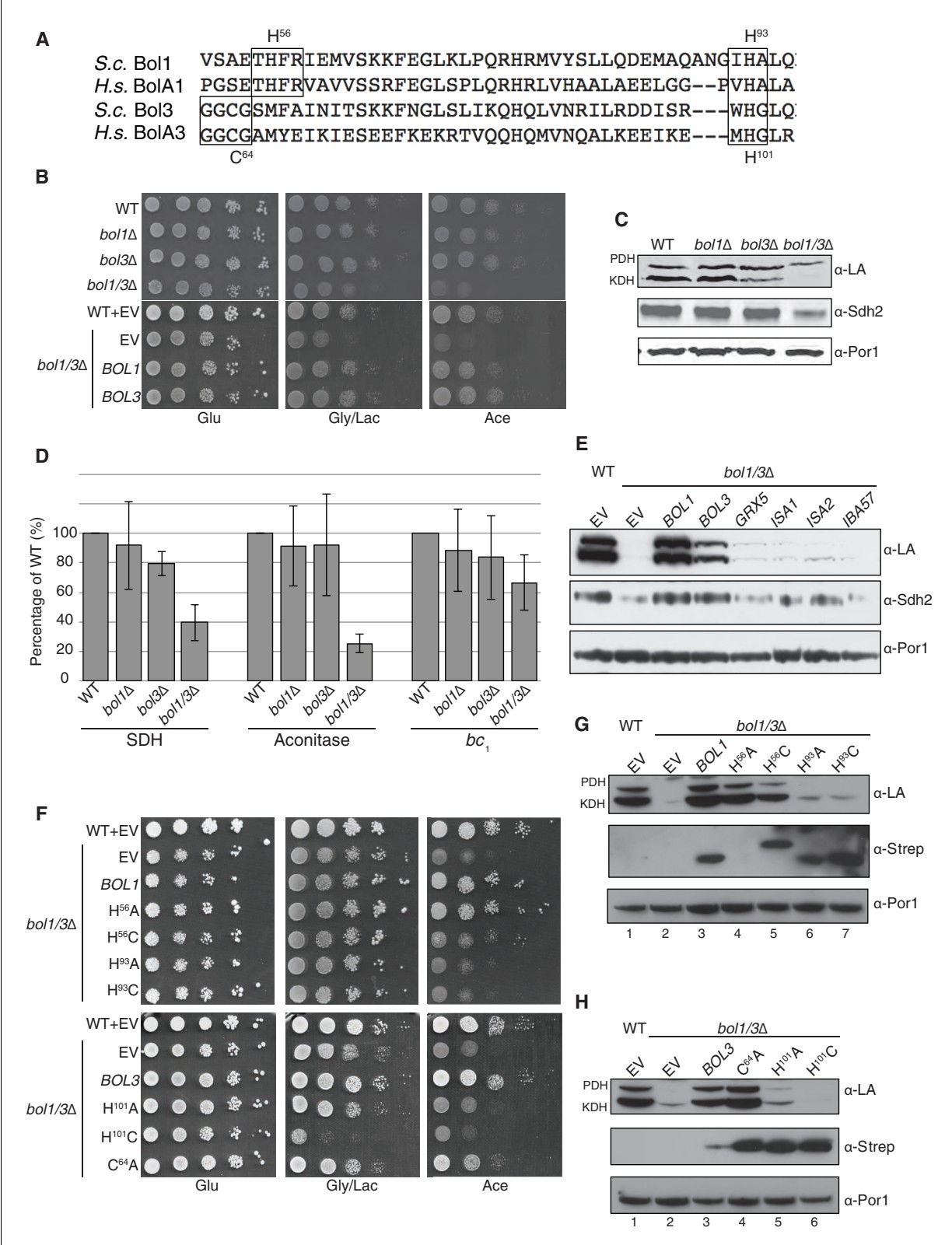

**Figure 4.** The Mitochondrial Bol1 and Bol3 proteins function in Fe-S biogenesis. Bol1 and Bol3 play roles in Fe-S cluster biogenesis in mitochondria (**A**) Partial sequences of yeast and human mitochondrial BolA proteins. Boxed are conserved motifs with proposed ISC ligands that were mutated in this work. (**B**) Respiratory growth defects of *bol1Δ* cells, *bol3Δ* cells and *bol1Δbol3Δ* double mutants and complementation by plasmid-borne *BOL1* or *BOL3*. (**C**) Steady-state levels of LA-conjugated proteins and Sdh2 in cells lacking Bol1 and/or Bol3. (**D**) Relative activity of SDH, cytochrome $bc_1$

*Figure 4 continued on next page*

*Figure 4 continued*

complex and aconitase were measured. Data are shown as mean ± SE (n=3). (E) Observation of LA moieties on PDH and KDH and Sdh2 steady-state levels by SDS-PAGE followed by immunoblotting in isolated mitochondria from *bol1/3Δ* cells over-expressing the indicated Fe-S cluster gene. (F) Respiratory function of Bol1 and Bol3 sequence variants in conserved residues were examined by yeast drop-test. All Bol1 variants were fused with a C-terminal Strep-tag and expressed on low-copy plasmids. All Bol3 variants were fused with a N-terminal Strep-tag between the MTS and the remainder of the protein and expressed on low-copy plasmids. (G and H) Steady-state levels of LA-conjugated proteins in cells lacking Bol1 and Bol3 with Bol1 variants (G) and Bol3 variants (H) exogenously expressed.

The following figure supplement is available for figure 4:

**Figure supplement 1.** Yeast growth tests evaluating the viability of cells expressing mitochondrial Bol1 and Bol3 N-terminal ligands mutated to lysine in the *bol1/3Δ* background.

contrast to cells harboring the Nfu1 $G^{194}C$ patient mutation, *nfu1Δ* cells have normal $F_1F_0$ ATPase levels on BN-PAGE and Cox2 steady-state levels suggesting that mitochondrial translation is normal without Nfu1.

We tested whether the dominant negative effect arises from changes in interactions between Nfu1 and client proteins. We performed affinity purification of Nfu1-Strep on Strep-Tactin beads for the WT and mutant alleles. The loss-of-function AxxANfu1mutant failed to show a detectable interaction with Aco1 (*Figure 3G*, lane 8) and was impaired in its interaction with Lys4 (*Figure 3—figure supplement 2*). In contrast, the $G^{194}C$ Nfu1mutant exhibited an enhanced interaction with Aco1 (*Figure 3G*, lane 10). An interaction with Sdh2 is unclear, since Sdh2 levels are markedly depleted in $G^{194}C$ Nfu1 cells. These data show the functional importance of the NfuC domain and its CxxC motif.

## The two mitochondrial BolA proteins function in Fe-S protein biogenesis

MMDS2 patients have been reported to have mutations in the mitochondrial BOLA3 protein (*Seyda et al., 2001*; *Cameron et al., 2011*; *Baker et al., 2014*). The clinical phenotypes of patients with mutations in *NFU1* or *BOLA3* were similar with neurological regression, infantile encephalopathy and hyperglycinemia (*Cameron et al., 2011*; *Navarro-Sastre et al., 2011*). In addition, biochemical defects in protein lipolation and succinate dehydrogenase were observed.

Due to the clinical and biochemical similarities in mutant NFU1 or BOLA3 patients, we tested the function of the yeast BOLA3 homolog, Bol3, and the related Bol1 protein (*Figure 4A*). In human cells, BOLA1 and BOLA3 are known to be mitochondrial proteins (*Willems et al., 2013*). We confirmed that Bol1 and Bol3 are likewise localized within the mitochondria of yeast cells (data not shown). Yeast devoid of either Bol1 or Bol3 lacks a clear respiratory phenotype, but a double *bol1Δbol3Δ* null strain displayed a growth defect on acetate medium and to a lesser extent on glycerol/lactate medium (*Figure 4B*). Mitochondria isolated from single mutants and the double null mutant were used for biochemical characterization studies. As with *nfu1Δ* cells, protein lipoylation was partially impaired in KDH in the *bol3Δ* null, but the defect in KDH lipoylation was enhanced in the *bol1Δbol3Δ* null strain (*Figure 4C*). SDH and aconitase activities were depressed in the double null strain, but not significantly changed in the individual single mutants (*Figure 4D*). The attenuation of aconitase activity in both *bol1Δbol3Δ* null and *nfu1Δ* cells is in contrast to BOLA3 and NFU1 patient mutant cells. A modest attenuation was seen in $bc_1$ activity in the *bol1Δbol3Δ* null strain, but this was not observed in *nfu1Δ* cells.

Since the respiratory growth defect of *nfu1Δ* cells was partially suppressed by overexpression of *ISA2*, we tested whether overexpression of a series of late mitochondrial Fe-S cluster assembly genes would likewise rescue the respiratory defect of *bol1Δbol3Δ* cells. No growth restoration was observed on acetate or glycerol/lactate media, and only minimal lipoylation of PDH and KDH was observed in cells harboring elevated levels of Grx5, Isa1, and Isa2 (*Figure 4E*).

BolA proteins are implicated in binding Fe-S clusters. Whereas Nfu1 is known to bind a [4Fe-4S] cluster at the homodimer interface, BolA proteins have been shown to bind [2Fe-2S] clusters in association with glutaredoxins as heterodimers (*Li and Outten, 2012*; *Roret et al., 2014*; *Li et al., 2012*). We evaluated the roles of potential Fe-S cluster ligands in Bol1 and Bol3. Bol1 has conserved His56 and His93 residues (*Figure 4A*), which in the case of *Arabidopsis thaliana* BolA1 the

corresponding His residues are apparent ligands to a [2Fe-2S] cluster in association with a monothiol glutaredoxin (*Roret et al., 2014*). Bol3 has conserved Cys64 and His101 residues in corresponding loops to that of Bol1 and are expected to serve as ligands for a Fe-S cluster. We replaced the conserved histidine residues with alanines or cysteines and tested phenotypic effects. We observed that the C-terminal His in each BolA protein was important for the respiratory growth of cells (*Figure 4F, G and H*). Whereas the $H^{101}A$ Bol3 mutant was non-functional, the variant containing a $H^{101}C$ substitution exhibited a synthetic sick phenotype in that the respiratory growth on glycerol/lactate medium was more impaired relative to the starting *bol1Δbol3Δ* null strain (*Figure 4F and H* lane 6). The Bol3 $C^{64}A$ mutant was only a partial loss-of-function allele. In contrast, the Bol1 $H^{93}A$ or $H^{93}C$ mutants exhibited similar loss-of-function phenotypes without any observed dominant negative effects. The upstream Bol1 $H^{56}A$ mutant retains function, but the $H^{56}C$ allele was a partial loss of function mutant (*Figure 4F and G*, lane 5).

Since substitutions in the upstream conserved His56 in Bol1 and Cys64 in Bol3 failed to yield a significant phenotype, they may not contribute to a candidate FeS cluster binding. To confirm this prediction, we converting the His56 in Bol1 and Cys64 in Bol3 to lysine residues to create electrostatic repulsion to a candidate FeS cluster Fe atom. Bol1 $H^{56}K$ and Bol3 $C^{64}K$ mutants did not exhibit an enhanced phenotype (*Figure 4—figure supplement 1*) ruling out that they are important FeS cluster ligands. Together, these data show a functional importance of Bol1 and Bol3 in mitochondrial Fe-S cluster biogenesis and highlights the need for the C-terminal conserved His in each protein for physiological function. Bol1 and Bol3, like Nfu1, are not essential for mitochondrial Fe-S protein biogenesis, as a bypass exists enabling limited respiratory growth on glycerol/lactate medium.

## Nfu1 and Bol3 physically interact with [4Fe-4S] client mitochondrial proteins

To glean further insights into the function of Nfu1, Bol1 and Bol3 in mitochondrial Fe-S cluster biogenesis, we performed proteomic analyses on affinity purified Nfu1, Bol1 and Bol3 proteins with each expressed as Strep fusions. Purification of each protein was accomplished on Strep-Tactin resin and protein eluates were analyzed by mass spectrometry. Multiple independent proteomic analyses were conducted on WT proteins as well as mutant proteins of each (G/T>H Nfu1, $H^{93}C$ Bol1 and $H^{101}C$ Bol3) (*Figure 5A and B*; *Figure 5—source data 1*). Of the mutant proteins, BolA3 $H^{101}C$ was synthetic sick in the *bol1Δbol3Δ* null strain (*Figure 4F*); Nfu1 G/T>H mimicked the severe dominant negative mutant, $G^{194}C$ found in patients, however the substitutions were less detrimental to growth (*Figure 3D*); and the Bol1 $H^{93}C$ variant was a loss-of-function mutant without a dominant negative characteristic (*Figure 4F*). Inspection of datasets of protein interactors revealed a common set of [4Fe-4S] client proteins associating with both Nfu1 and Bol3. These include Aco1, Aco2, Lys4, Sdh2, Lip5 and Bio2. For all client proteins except Sdh2, the observed total spectral count that was markedly higher for clients purified with mutant Nfu1 and Bol3 variants (*Figure 5A* and *Figure 5—source data 1*). Additionally, the mutant forms of Bol3 and Nfu1 both co-purified with the ISA complex component, Isa2 (*Figure 5B* and *Figure 5—source data 1*). The physical interactions of Nfu1 with the clients, Aco1, Lys4, Aco2 and Sdh2, and with the ISA complex are consistent with the results shown by affinity purification experiments followed by SDS-PAGE and immunoblotting (*Figure 1F,G and H*). Two Bol3 pulldown studies revealed limited levels of copurified Nfu1, although we never observed Bol3 in the pulldown of Nfu1 (*Figure 5—source data 1*).

Unlike Bol3, Bol1 purification did not lead to appreciable co-purification of [4Fe-4S] client proteins, but Grx5 was isolated as a reproducible interactor with WT but not the loss-of-function $H^{93}C$ Bol1 mutant (*Figure 5B*). Grx5 was a significantly less abundant interactor with Bol3 or Nfu1. Human BOLA1 was previously shown to associate with Grx5 in HEK293 cells (*Willems et al., 2013*).

We conducted in vitro experiments to verify the observed selective interaction of Bol1 with Grx5 using human orthologs. Solution binding analyses were conducted using microscale thermophoresis, which assesses molecular diffusion in a microscopic temperature gradient. Due to their higher stability, we used the human proteins. Apo-GLRX5 or the holo-GLRX5 dimer containing the bridging [2Fe-2S] cluster were incubated with either recombinant human BOLA1 or BOLA3 proteins for assessment binding (*Figure 5C* and *Figure 5—figure supplement 1*). Holo-GLRX5 associated with BOLA1 with a 50-fold greater affinity than with BOLA3, while similar affinities of apo-GLRX5 were observed for the two human BOLA proteins. As a control, no significant interaction of the BOLA proteins with the human [2Fe-2S] ferredoxin FDX2 was observed. Together, these results indicate a strong preference

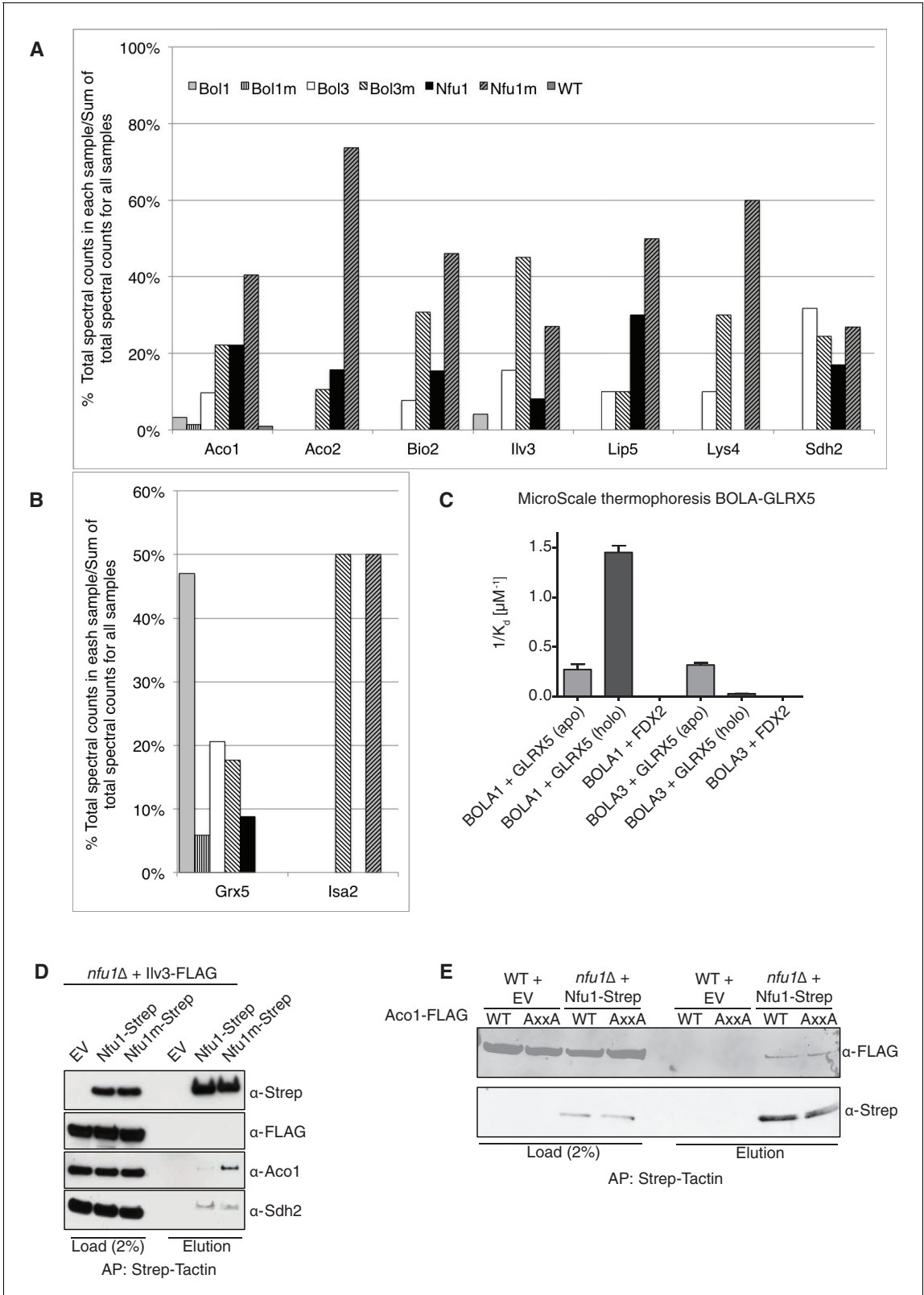

**Figure 5.** Proteomic analysis of Nfu1, Bol1 and Bol3 establishes function within mitochondrial Fe-S for Bol1 and Bol3. (**A** and **B**) Percentages of spectral counts identified by MS proteomics. Percentages were calculated by the number of spectral counts identified for a denoted protein in an individual Strep-tagged protein divided by the total number of spectral counts for that protein identified from all seven samples. Strep-tagged proteins were expressed from low-copy plasmids in corresponding single deletion mutants. Samples were Strep-affinity purified as in *Figure 3*. Bol1m is the H93C

*Figure 5 continued on next page*

*Figure 5 continued*

variant. Bol3m is the H$^{101}$C variant. Nfu1m is the G/T>H variant. WT is wild-type BY4741 expressing an empty vector. All were fused with a C-terminal Strep-tag. WT is BY4741 wild type harboring a low-copy empty plasmid. (**C**) Human GLRX5 or NFU1 were used in apo- and holo- form and mixed at increasing concentrations with 200 nM fluorescently labelled BOLA1 or BOLA3. Microscale thermophoresis were performed and dissociation constants (K$_d$) were determined. Error bars indicate the SD (n=3). (**D**) Strep-tag affinity purification of Nfu1-Strep in the presence of ectopically expressed Ilv3-FLAG. (**E**) Affinity purification using Strep-Tactin agarose beads to purify Nfu1-Strep from an *nfu1Δ* background expressing either WT Aco1 or Aco1 AxxA mutant.

The following source data and figure supplements are available for figure 5:

**Source data 1.** (Table 1) Spectral counts, unique peptides, and coverage of mitochondrial Fe-S client proteins, bait proteins, and Fe-S assembly machinery identified by MS proteomics.

**Figure supplement 1.** Interaction studies of human BOLA proteins with GLRX5.

**Figure supplement 2.** Ilv3 activity assay using wild-type and *nfu1Δ* purified mitochondria along with wild-type overexpressing Ilv3 as a control.

of BOLA1 for the holoform of GLRX5. We predict, based on the yeast Bol1 H$^{93}$C mutant, that BOLA1:GLRX5 interaction is mediated by a [2Fe-2S] cluster.

The Nfu1 and Bol3 proteomics experiments did not identify any novel mitochondrial [4Fe-4S] cluster client proteins. Interestingly, the [2Fe-2S] enzyme dihydroxyacid dehydratase (Ilv3) was recovered in multiple independent mass spectrometry analyses in Nfu1 and Bol3 samples. However, we were unable to verify that interaction when using a FLAG-tagged Ilv3 chimera in the Nfu1-Strep affinity capture (*Figure 5D*). Furthermore, enzymatic activity of Ilv3 was not altered in *nfu1Δ* cells (*Figure 5—figure supplement 2*). Thus, Nfu1 does not appear to be important for the function of the [2Fe-2S] Ilv3 enzyme.

We sought to address whether the binding of Nfu1 to a client protein was mediated through a bridging [4Fe-4S] cluster. Aco1 binds its [4Fe-4S] cluster through a conserved CxxC motif and one distant Cys residue in the primary sequence. We generated a double AxxA mutant and tested its ability to associate with Nfu1-Strep. No difference in binding was observed between WT and the AxxA Aco1 proteins with Nfu1 (*Figure 5E*).

## Nfu1 and Bol3 function together in [4Fe-4S] cluster transfer from the ISA complex to apo-client proteins

The distinct overlap of [4Fe-4S] client protein interactors between Bol3 and Nfu1 suggested a potential overlap or partnership in the function of the two proteins in late step [4Fe-4S] cluster transfer. We tested whether a genetic linkage exists between the proteins by evaluating whether a synthetic phenotype exists in cells lacking Bol1, Bol3 and Nfu1. The triple deletion cell (*bol1Δbol3Δnfu1Δ*, designated *bΔΔnfu1Δ*) exhibited a strong synergistic growth defect on glycerol/lactate medium (*Figure 6A*). While the defects are too severe to see the synergism by protein lipoylation and Sdh2 steady-state levels, the enzymatic activities of SDH and aconitase do reflect a synergistic effect (*Figure 6B and C*). In addition, the level of assembled F1F0 ATPase was markedly reduced in the triple mutant (*Figure 6D*). The severity of the phenotype and the impairment in F1F0 ATPase in the triple mutant likely arises from reduced mitochondrial translation likely through RNaseP, similar to the dominant negative Nfu1 G>C mutant discussed above (*Figure 3F*). The growth defect of the triple mutant can be partially rescued by re-expression of *BOL1* or *NFU1*, but not by *BOL3* (*Figure 6E*). This may suggest that Bol3 requires Nfu1 for its function.

Affinity purification of Nfu1-Strep expressed in the *bol1Δbol3Δnfu1Δ* triple null mutant was carried out to test the effect of loss of the two BolA proteins on the interaction of Nfu1 with [4Fe-4S] client proteins. As can be seen in *Figure 6F* there was enhanced co-purification of Aco1 in the absence of Bol1 and Bol3. Likewise, a similarly enhanced interaction between client proteins and Nfu1 was apparent in cells lacking a functional ISA complex in *isa2Δ* cells (*Figure 6G*). These data are consistent with a role of Nfu1 in [4Fe-4S] cluster transfer from the ISA complex to client proteins.

Given the strong genetic interaction between the mitochondrial BOLA genes and *NFU1*, we attempted to substantiate the linkage. The proteomic results suggested an association of Bol1 and

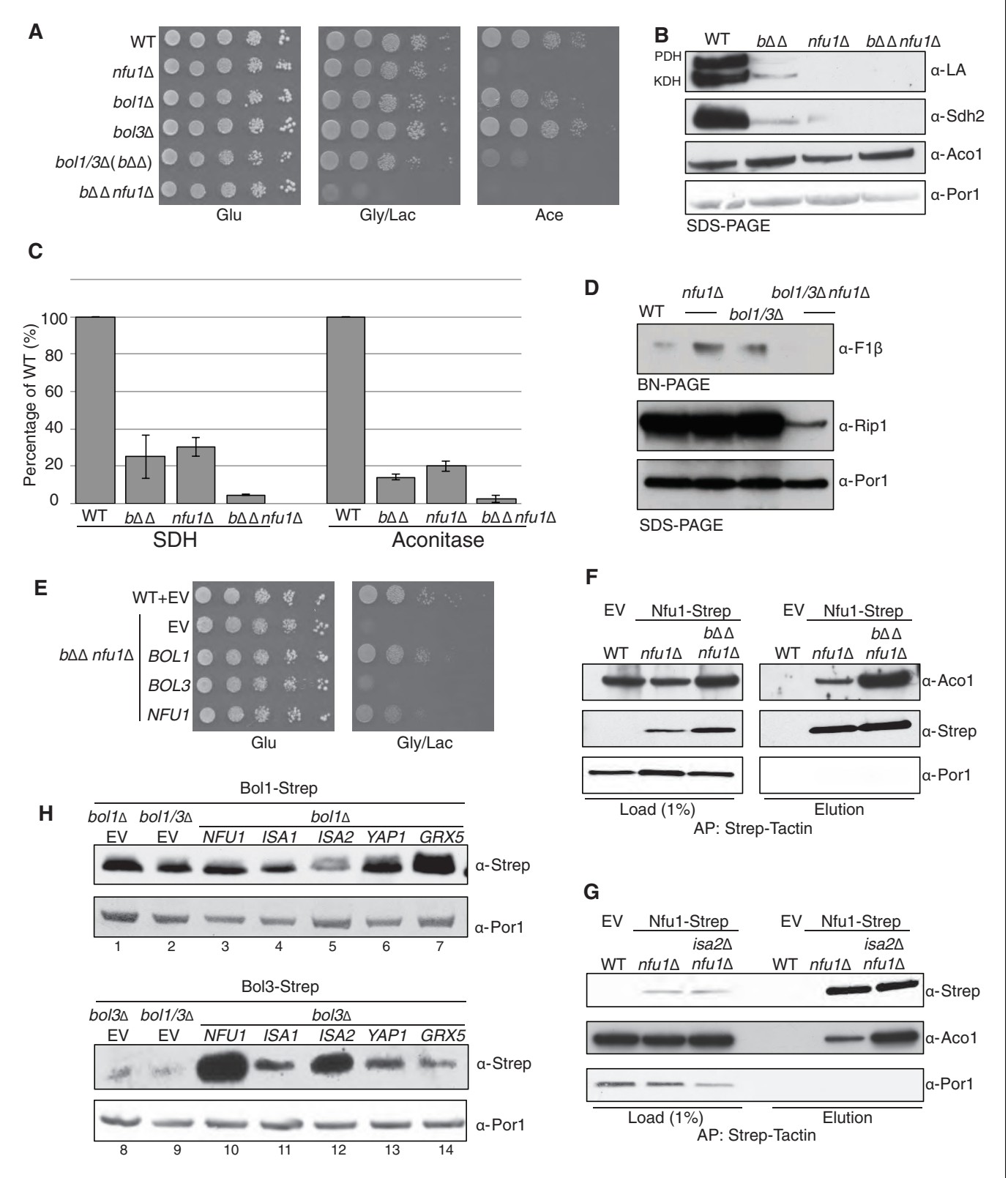

**Figure 6.** Nfu1 and Bol3 function together in [4Fe-4S] delivery. (A) Exacerbated respiratory growth defects of *bol1Δbol3Δnfu1Δ* triple mutants (designated *bΔΔnfu1Δ*) compared to *nfu1Δ* single mutants and *bol1Δbol3Δ* double mutants on non-fermentable carbon sources. (B) Steady-state levels of LA-conjugated proteins and Sdh2 in the absence of Bol1, Bol3 or Nfu1. (C) Relative activity of SDH and aconitase in the absence of Bol1, Bol3 or Nfu1. Data are shown as mean ± SE (n=3) (D) BN-PAGE and SDS-PAGE analysis of [4Fe-4S] cluster independent enzymes in the *bΔΔnfu1Δ* triple

*Figure 6 continued on next page*

*Figure 6 continued*

deletion mutant background. F1β is a subunit of ATP synthase. (E) Respiratory growth of *bΔΔnfu1Δ* triple mutants harboring plasmid-borne *BOL1, BOL3* and *NFU1*, respectively. (F) Strep-tag purification of Nfu1-Strep in the absence of Bol1 and Bol3. (G) Strep-tag purification of Nfu1-Strep in the absence of Isa2. (H) Steady-state levels of Bol1-Strep (upper panel) and Bol3-Strep (bottom panel) in response to overexpression of genes as indicated.
The following figure supplement is available for figure 6:

**Figure supplement 1.** SDS-PAGE followed by immunoblotting to evaluate the different steady state levels of Nfu1-Strep, Bol1-Strep, and Bol3-Strep while being expressed under the same heterologous *MET25* promoter and *CYC1* terminator.

Grx5, whereas Bol3 was associated with Nfu1 function. Mitochondrial BolA proteins are low abundance molecules (*Figure 6—figure supplement 1*) making co-immunoprecipitation studies challenging. Because of this, we tested whether increasing the levels of candidate interacting proteins would alter the abundance of Bol1 or Bol3. As can be seen in *Figure 6H*, the steady-state levels of Bol3, but not Bol1, were dramatically increased in cells with elevated levels of Nfu1. Additionally, *ISA1* and *ISA2* overexpression resulted in a modest increase in Bol3, but not Bol1, protein levels. In contrast, Grx5 overexpression led to a marked enhancement in Bol1 levels without altering Bol3 (*Figure 6H*). In these studies Strep-chimeras of Bol1 and Bol3 were expressed from heterologous promoters, so the changes in protein levels are likely occurring through post-transcriptional stabilization. These stabilization experiments corroborate the genetic and proteomic experiments, all of which suggest that Bol3 (BOLA3) functions with Nfu1 in [4Fe-4S] cluster transfer to client proteins and Bol1 functions with Grx5 for a yet to be determined purpose.

## Nfu1 exists as a steady-state homodimer lacking Bol3

Mitochondrial lysates were subjected to gel filtration studies to assess the extent of interaction between Nfu1 and Bol3. Mitochondrial lysates prepared from either Nfu1-Strep or Bol3-Strep cells were separately chromatographed and fractions were assayed for Nfu1 or Bol3. The bulk of Nfu1 eluted in fractions corresponding to a globular mass of ~47 kDa, consistent with a homo-dimeric complex (*Figure 7A*). Nfu2 from *Arabidopsis* is a dimeric species both as an apo-protein and with a [4Fe-4S] cluster (*Gao et al., 2013*). In contrast, Bol3 eluted predominantly in fractions corresponding to a globular mass of 13 kDa consistent with a monomeric protein. No significant co-elution was observed between Nfu1 and Bol3, indicating that any Nfu1/Bol3 interaction is transient in nature. An additional set of chromatographic studies were done with mitochondrial lysates containing Nfu1-Strep in which lysates were treated with either 0.1 mM DTT or 2 mM dithionite to assess whether the apparent Nfu1 dimer was a disulfide-linked homo-dimer or a Fe-S cluster bridged complex (*Figure 7B*). The abundance of Nfu1 in fraction 13 assessed by immunoblotting indicated that the elution properties of Nfu1 were unaffected by preincubation with DTT (followed by alkylation of cysteines by Iodoacetamide), whereas treatment with dithionite attenuated the apparent dimeric complex abundance. Fe-S clusters are susceptible of disassembly with dithionite treatment, suggesting that a significant fraction of steady-state Nfu1 in WT yeast mitochondria may be Fe-S loaded. In support of this conclusion is the observation that the Nfu1 AxxA mutant fractionates predominantly as a monomer.

If Nfu1 exists in a FeS-loaded conformer, the question arose whether [4Fe-4S]-Nfu1 serves as a reservoir of [4Fe-4S] clusters for client proteins. We specifically focused on Lip5 that catalyzes formation of lipoic acid. In its catalytic cycle, one of its [4Fe-4S] clusters is consumed to provide two sulfur atoms needed to generate lipoic acid (*Cicchillo and Booker, 2005*; *Cicchillo et al., 2004*). Thus, [4Fe-4S] cluster regeneration is needed to support Lip5 catalysis and lipoic acid levels. To address a role of Nfu1 in cluster regeneration in Lip5, we utilized cells containing a chromosomal *NFS1* under the control of *GAL1* promoter enabling glucose-mediated repression of Nfs1 expression. The GAL-*NFS1* strain was transformed with an empty vector or a vector containing either *NFU1* under the control of the regulatable *MET25* promoter, *BOL3* or *ACO1*. If Nfu1 were a reservoir of [4Fe-4S] clusters, we predicted that elevated levels of holo-Nfu1 would enable sufficient [4Fe-4S] cluster transfer to Lip5 to support lipoic acid formation in cells depleted of Nfs1. Cells pre-cultured in galactose were shifted to glucose-containing medium to repress *NFS1* expression and mitochondrial lysates were collected 8 hr later (*Figure 7C*). The lipoic acid level in pyruvate dehydrogenase was

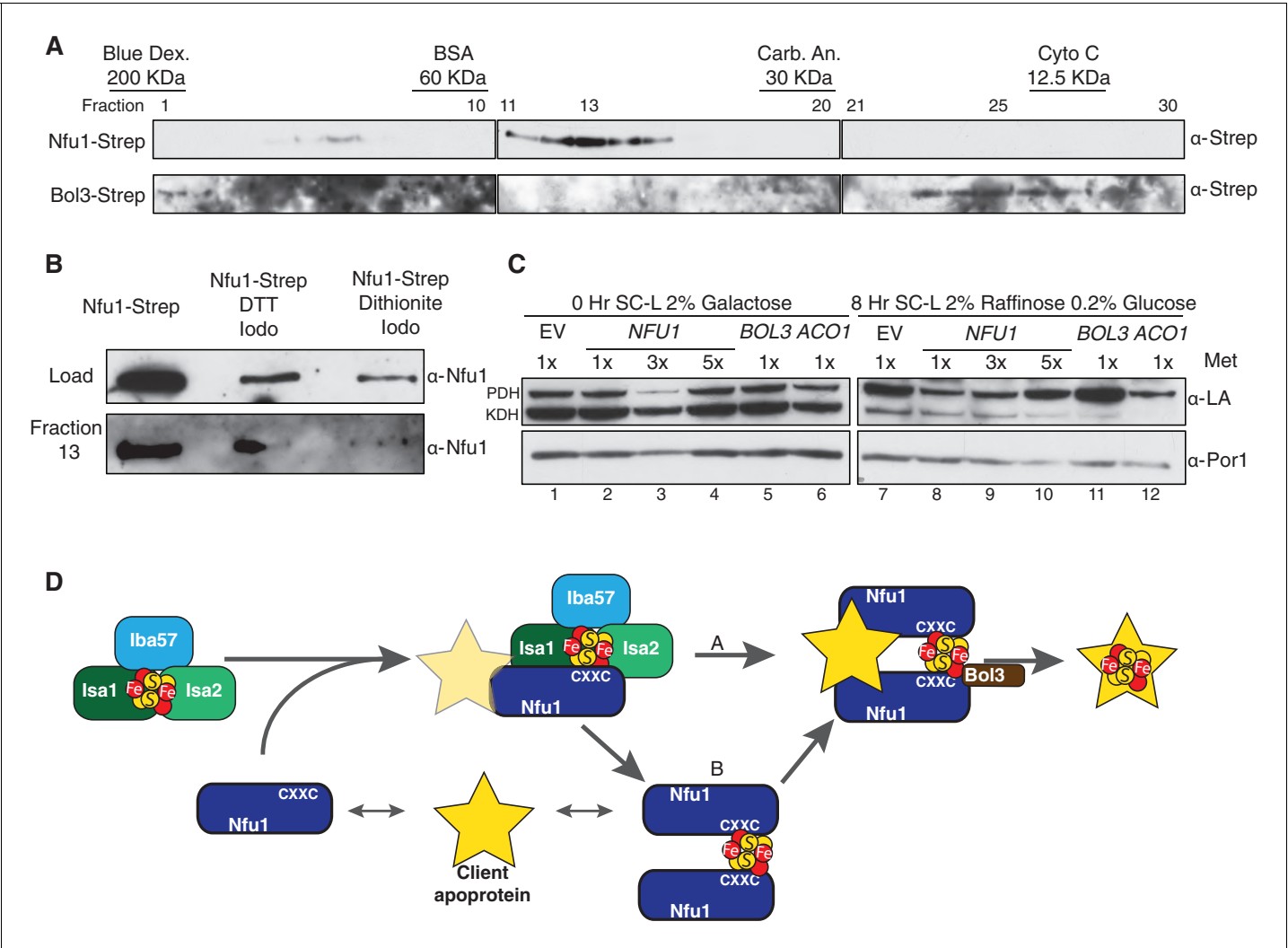

**Figure 7.** Nfu1 exists as a dimer bridged by a Fe-S cluster. (**A**) Immunoblotting of fractions from *nfu1Δ* + Nfu1-Strep or *bol3Δ* + Bol3-Strep lysates separated by size exclusion chromatography. Molecular weight standards [bovine serum albumin (BSA), carbonic anhydrase (CA) and cytochrome c] are displayed above the corresponding fractions were used to create a standard curve to calculate apparent molecular weights. Fraction 13 has an apparent molecular weight of 47.6 KDa and Fraction 26 has an apparent molecular weight of 13.2 kDa. (**B**) Immunoblotting of fraction 13 from *nfu1Δ* + Nfu1-Strep lysates pretreated with nothing, 100 mM DTT followed by 200 mM iodoacetamide, or 2 mM dithionite followed by 200 mM iodoacetamide were separated by size exclusion chromatography. (**C**) GAL-*NFS1* shutdown was induced over 8 hr with over-expression of *NFU1*, *BOL3*, and *ACO1* and LA levels were observed by immunoblotting. Nfu1 protein levels were reduced with increasing amounts of methionine by utilizing a *MET25* promoter is repressed with excess methionine (1x = 0.6 mM methionine). (**D**) A working model of late stage mitochondrial iron-sulfur cluster biogenesis and delivery. Two potential pathways of cluster transfer are shown in **A** and **B**.

reduced, rather than maintained, in cells expressing elevated levels of Nfu1 (low 1x methionine cultures) (*Figure 7C*, lane 8, top band), but unaffected in cells overexpressing Bol3 (lane 11). Overexpression of the aconitase mimicked the reduced lipoic acid levels seen with elevated Nfu1 (lane 12). These results do not support a reservoir function of [4Fe-4S]-Nfu1 and suggest that elevated levels of a [4Fe-4S] protein, e.g. [4Fe-4S] Nfu1, may negatively impair routing [4Fe-4S] transfer to Lip5 for lipoic acid formation.

## Discussion

A role of Nfu1 in Fe-S cluster biogenesis has long been implicated (*Jacobson et al., 1989*; *Schilke et al., 1999*); however, its molecular mechanism has not been definitely established. Patients

harboring mutations in NFU1, as well as BOLA3, exhibit biochemical abnormalities in a set of [4Fe-4S] enzymes leading to speculation that Nfu1, and BolA3, function as a late Fe-S maturation factor (*Navarro-Sastre et al., 2011*; *Py et al., 2012*) or that Nfu1 is an alternate Fe-S cluster synthesis scaffold protein used for a subset of specific Fe-S client proteins (*Cameron et al., 2011*; *Tong et al., 2003*). The phenotypic similarity between Nfu1 and BolA3 mutations suggests the two proteins function in a common step of the Fe-S protein maturation pathway.

We demonstrate in studies using yeast as a model system that the yeast orthologs of human NFU1 and BOLA3 function in a late step of transfer of [4Fe-4S] clusters to specific client proteins. Yeast lacking Nfu1 are partially deficient in the [4Fe-4S] enzymes aconitase, succinate dehydrogenase and lipoic acid synthase. The defect in lipoic acid synthase is highlighted by the pronounced defect in protein lipoylation in mitochondria. The defect in yeast lacking Bol3 is modest, but is exacerbated in cells lacking both mitochondrial Bol3 and Bol1. The double null cells show related partial defects in [4Fe-4S] enzymes aconitase, succinate dehydrogenase and lipoic acid synthase, although the defects are not as pronounced as in *nfu1Δ* cells. Yeast lacking all three proteins Nfu1, Bol1 and Bol3 show an exaggerated phenotype approaching the defect seen in cells lacking the ISA complex required for [4Fe-4S] cluster synthesis. Clearly, *nfu1Δ* cells do not exhibit any defects in enzymes dependent on [2Fe-2S] centers, suggesting that Nfu1 functions in the [4Fe-4S] cluster transfer pathway.

Our systematic approach to identify endogenous binding partners of Nfu1, Bol1 and Bol3 revealed the step in Fe-S cluster biogenesis in which they function. Affinity purification of Nful1 coupled with mass spectrometry led to the identification of [4Fe-4S] client proteins as physically associating proteins of Nfu1. It is of interest that the $G^{194}C$ Nfu1 variant exhibiting a partial dominant negative effect showed enhanced interaction with the same client proteins. This yeast mutant mimics the known $G^{208}C$ patient mutation in human NFU1 that causes MMSD. Recombinant Nfu1 has been shown to bind a [4Fe-4S] cluster and $^{55}Fe$ in vivo labeling studies showed a strong increase in $^{55}Fe$ binding by the patient mimic $G^{194}C$ Nfu1 yeast variant (*Navarro-Sastre et al., 2011*). Moreover, it is also noteworthy that Gly194 is in juxtaposition to the CxxC motif, which has been shown to bind Fe-S clusters. Therefore, it is plausible that the dominant negative effect of the $G^{194}C$ Nfu1 variant may result from the inefficient release of [4Fe-4S] clusters from the Nfu1 variant to client proteins.

The dramatic phenotype of cells harboring $G^{194}C$ Nfu1 is likely due to secondary effects of impaired lipoic acid formation. As mentioned, yeast lacking enzymes involved in octanoic acid formation or lipoic acid synthase are deficient in tRNA processing by RNase P leading to attenuation in mitochondrial translation (*Schonauer et al., 2008*; *Hiltunen et al., 2009*). Consistent with impaired RNase P function, $G^{194}C$ Nfu1 cells are markedly attenuated in levels of the $F_1F_0$ ATPase and Cox2 steady-state levels. In contrast, the RNase P function is normal in either *nfu1Δ* cells or *bol1Δbol3Δ* cells based on normal $F_1F_0$ ATPase assembled complexes.

The physical interactions of Nfu1 with Isa1 and Isa2 corroborate our model that Nfu1 functions in [4Fe-4S] cluster transfer to client proteins. Interestingly, we isolated Isa2 as a suppressor of the respiratory defect of *nfu1Δ* cells. Whereas the condensation of two [2Fe-2S] to form a single [4Fe-4S] cluster requires the participation of Isa1, Isa2 and Iba57, Isa2 is capable of forming homo-dimers that may exert a limited transfer function as proposed for Nfu1.

The same [4Fe-4S] client proteins were pulled down in affinity purification of Bol3, but not Bol1, compared to proteins interacting with Nfu1. In the case of Bol3, the dominant negative $H^{101}C$ Bol3 variant also showed enhanced interactions with [4Fe-4S] client proteins. The dominant negative phenotype of the $H^{101}C$ Bol3 mutant (putative Fe-S ligand) but only loss of function phenotype for the $H^{101}A$ mutant is consistent with a model that Bol3 His101 participates in Fe-S cluster transfer.

The partial deficiency of [4Fe-4S] enzyme activities in *nfu1Δ* cells suggests that the function of Nfu1 may be conditionally important in [4Fe-4S] cluster transfer and that a bypass mechanism exists in yeast. We demonstrate that Nfu1 in yeast has a heightened importance in cells undergoing oxidative metabolism as opposed to anoxic metabolism. In addition, *nfu1Δ* cell growth defect is partially suppressed with supplemental GSH in the growth medium. Identification of the Yap1 and Yap2, transcription factors that are important for oxidative stress tolerance, as high copy suppressors emphasized the importance of Nfu1 during oxidative metabolism.

One curiosity is that human patients with mutations in NFU1 or BOLA3 lack defects in mitochondrial aconitase, whereas the yeast mutants, *nfu1Δ* and the double *bol1Δ bol3Δ*, exhibit a partial aconitase defect. There are two implications of this result. First, Nfu1 may exhibit different client

selectivity in the actual transfer of [4Fe-4S] clusters. Although Nfu1 binds many [4Fe-4S] client proteins, it may facilitate cluster transfer to select clients and this may differ between human and yeast cells as in the case of aconitase. This postulate is supported by the observed role for Nfu1 in Aco1 and Lip5 activation, but not the function of Aco2 and Lys4. Second, since the partial respiratory function persists in *nfu1Δ* cells, Nfu1 may facilitate cluster transfer in oxidative growth conditions and this may differ between yeast and human cells.

One dramatic phenotype in human and yeast Nfu1 mutant cells is impaired protein lipoylation. Yeast and human cells require lipoylation on E2 subunits of pyruvate dehydrogenase, 2-oxoglutarate dehydrogenase and the glycine cleavage enzyme complex. In addition, the human branched chain 2-oxoacid dehydrogenase requires lipoylation for function. Lip5 catalyzing formation of the lipoate coenzyme binds two [4Fe-4S] clusters, one of which serves as the sulfur donor for lipoic acid formation in a radical S-adenosylmethionine dependent reaction (*Cicchillo and Booker, 2005*; *Cicchillo et al., 2004*). Two sulfide ions from this auxiliary cluster are used for formation of lipoate resulting in disassembly of the cluster. Each catalytic cycle of the enzyme requires repair or replacement of the auxiliary cluster (*Cronan, 2014*). Nfu1 may have a specialized role in cluster repair in lipoic acid synthase or alternatively provides a [4Fe-4S] replacement.

For most [4Fe-4S] client proteins, Nfu1 appears to have evolved to shield its [4Fe-4S] cluster from endogenous oxidants during the cluster transfer step. Oxidants are generated by 2-oxoacid dehydrogenases (*Boutigny et al., 2013*), so Nfu1-mediated cluster transfer may be critical to ensure intact [4Fe-4S] insertion. Nfu1 may also serve as a chaperone of apo-client protein, preventing their aggregation in the absence of a bound Fe-S cluster. Additional studies are necessary to define these candidate roles.

Bol3, but not Bol1, was found to associate with [4Fe-4S] client proteins, whereas Bol1 reproducibly associated with Grx5 both in in vivo and in vitro studies. BolA:glutaredoxin complexes reported to date only bind [2Fe-2S] clusters (*Li and Outten, 2012*). Thus, Bol1 is anticipated to function with Grx5 in a [2Fe-2S] cluster step, whereas Bol3 is likely to function, independent of Grx5, in a Nfu1-mediated [4Fe-4S] cluster step. These studies suggest that Bol1 and Bol3 have specialized functions within the same pathway, such that cells lacking both Bol1 and Bol3 have a synthetic defect.

In summary, the present work suggests that Nfu1 has a significant role in a late step transfer of [4Fe-4S] clusters to select client proteins. Nfu1 binds the client proteins independent of the ISA complex and its association with the ISA complex may serve to recruit apo-clients to the ISA complex where [4Fe-4S] clusters are formed (*Figure 7D*). Some [4Fe-4S] client proteins may get their [4Fe-4S] cluster directly from the ISA complex, whereas others may derive their clusters after prior transfer of a [4Fe-4S] cluster to Nfu1. In these cases Nfu1 facilitates the process as an adapter protein in oxidatively growing cells. Additional work is required to discern the client selectivity in [4Fe-4S] cluster transfer by Nfu1. This model of eukaryotic Nfu1 function resembles the role of the *E. coli* Nfu1 ortholog NfuA, which binds a subset of Fe-S apo-client proteins and facilitates cluster transfer especially under oxidative stress conditions (*Py et al., 2012*; *Angelini et al., 2008*; *Boutigny et al., 2013*). Likewise, the *Azobacter* NfuA is reported to be critical under oxidative growth conditions (*Bandyopadhyay et al., 2008*). In the case of *E. coli*, NfuA cluster transfer is likely mediated directly by NfuA (*Py et al., 2012*). Bol3 likely functions with Nfu1 in cluster transfer, but its mechanism remains nebulous. Clearly, interaction studies separate Bol1 and Bol3 into two distinct classes, with Bol3 working with Nfu1 in [4Fe-4S] client binding and Bol1 working with Grx5, which has one known function upstream of the ISA complex (*Uzarska et al., 2013*; *Kim et al., 2010*; *Banci et al., 2014*). However, cells lacking both mitochondrial BolA proteins show a synthetic defect. The Bol3 protein may facilitate [4Fe-4S] cluster dissociation from either the ISA complex or Nfu1 in [4Fe-4S] cluster transfer. Additional work will be required to discern their mechanisms.

## Materials and methods

### Yeast strains and plasmids

BY4741 strains were used unless indicated otherwise. Deletion strains were generated by homologous recombination and confirmed by PCR analyses of loci as described earlier (*Longtine et al., 1998*). Plasmids used in this study were constructed using general subcloning techniques. For mutagenesis or adding epitope tags, Phusion DNA Polymerases (Thermo Fisher Scientific, Waltham, MA)

were used. All plasmid-borne genes were expressed under the *MET25* promoter and the *CYC1* terminator unless indicated otherwise.

## Strep-tag affinity purification

Affinity purifications of Strep-tagged proteins were conducted using Strep-Tactin superflow beads (Qiagen, Germany) following the manufacturer's instruction with slight changes. Briefly, isolated mitochondria were solubilized with 0.1% n-dodecyl maltoside (DDM) in the lysis buffer, 50 mM $NaH_2PO_4$ (pH 8.0), 300 mM NaCl and 1x protease inhibitor (cOmplete mini, Roche, Switzerland), for 30 min on ice. After clarification of solubilized mitochondria by high-speed centrifugation, the supernatants were incubated with Strep-Tactin superflow beads for 16 hr at 4°C. The beads were washed five times with the lysis buffer. Strep-tagged proteins bound to the beads were eluted with 2.5 mM dethiobiotin in the lysis buffer, which were subjected to mass spectrometry analyses or immunoblotting.

## Enzymatic activity assay

Activity assays for aconitase, succinate dehydrogenase (SDH), cytochrome $bc_1$ complex and cytochrome *c* oxidase were performed as described previously (*Atkinson et al., 2011*; *Na et al., 2014*). Aconitase activity was determined by measuring the initial rate of conversion of 100 mM *cis*-aconitate to isocitrate in 50 mM Tris (pH 7.4) at 240 nm. Soluble fractions of mitochondria were obtained by repetitive freeze-thaw. SDH activity was measured by quinone-mediated reduction of dichlorophenolindophenol (DCPIP) upon succinate oxidation at 600 nm. For cytochrome $bc_1$ complex activity, the reduction rate of cytochrome *c* was measured upon the oxidation of reduced decylubiquinol at 550 nm. Cytochrome *c* oxidase activity was determined by measuring the initial rate of oxidation of cytochrome *c* oxidation (*Pierrel et al., 2007*). Dihydroxy acid dehydratase (Ilv3) catalytic activity was assayed using an end point assay measuring 2,4-dinitrophenylhydrazine (DNPH) as a proton acceptor as described previously (*Limberg et al., 1995*). Purified mitochondria (30 μg) were lysed by sonication in assay buffer (20 mM $KPO_4$ and 10 mM $MgCl_2$), spun at 20,000 ×g for 15 min, before incubation with 100 mM dihydroxyisovalerate for 10 min in a total volume of 1 ml. The reaction was quenched with 100 μl of 50% TCA. Next 200 μl of DNPH (saturated in 2 N HCl) was added for 15 min when 500 μl of 2.5N NaOH was added to quench the reaction. The absorbance was measured at 540nm by a UV-VIS spectrophotometer.

## Mass spectrometry analysis

The purified Strep-tagged protein complexes were reduced, alkylated and digested as described (*Kaiser and Wohlschlegel, 2005*; *Wohlschlegel, 2009*). The digested peptide mixture was desalted using C18-packed pipette tips (Thermo Fisher Scientific) and fractionated online using a 75 μM inner diameter fritted fused silica capillary column with a 5 μM pulled electrospray tip and packed in-house with 15 cm of Luna C18 (2) 3 μM reversed phase particles. The gradient was delivered via an easy-nLC 1000 ultra high-pressure liquid chromatography (UHPLC) system (Thermo Fisher Scientific). MS/MS spectra were collected on a Q-Exactive mass spectrometer (Thermo Fisher Scientific) (*Kelstrup et al., 2012*; *Michalski et al., 2011*). Data analysis was carried out using the ProLuCID and DTASelect2 implemented in the Integrated Proteomics Pipeline - IP2 (Integrated Proteomics Applications, Inc., San Diego, CA) (*Cociorva et al., 2007*; *Tabb et al., 2002*; *Xu et al., 2006*). Protein and peptide identifications were filtered using DTASelect and required at least two unique peptides per protein with a peptide-level false positive rate of 5% as estimated by a decoy database strategy (*Elias and Gygi, 2007*). Normalized spectral abundance factor (NSAF) values were calculated as described (*Florens et al., 2006*) and multiplied by a factor of $10^5$ for readability.

## Size exclusion chromatography

Purified mitochondria (1.5 mg) were lysed by sonication in 50 mM $NaPO_4$ 150 mM NaCl (pH 7.0) buffer. Lysates were precleared and filtered prior being applied to a HiLoad Superdex 75 PG 16/600 column (GE Healthcare Life Sciences, United Kingdom) with a flow rate of 1 mL/Min. Fractions of 1.33 mL were collected, TCA precipitated and analyzed by immunoblotting.

## Affinity measurements using microscale thermophoresis (MST)

For MicroScale Thermophoresis the proteins were fluorescently labeled using the Monolith NT Protein Labeling Kit RED with NT-647 dye as recommended by the supplier (NanoTemper Technologies, Germany). The fluorescently labelled protein (200 nM) was titrated with serial dilutions of unlabeled protein (from 200 µM to 6.1 nM) in buffer containing 50 mM $KP_i$, pH 7.4, 150 mM NaCl, 5% glycerol, 0.05 mg/mL BSA, and 0.05% Tween20. Thermophoresis assays were performed using Monolith NT.115 at 21°C (LED power – between 40% and 60%, IR laser power 75%) in standard capillaries under anaerobic conditions. At least three independent experiments were recorded at 680 nm. The thermophoresis data were processed by Nano Temper Analysis 1.2.009 and GraphPad Prism5 software to estimate the $K_d$ values.

## Chemical reconstitution of Fe-S clusters

Chemical reconstitution was done in a COY (Grass Lake, MI) anaerobic chamber using freshly dissolved stock solutions. Protein solutions were reduced with 5 mM DTT for 2–3 hr on ice in reconstitution buffer (50 mM Tris-HCl, pH 8.0, 150 mM NaCl, 5% glycerol). Reconstitution was started at room temperature by the addition of a 2-3-fold excess of ferric ammonium citrate by inverting the tube. After 5 min a 2-3-fold excess of lithium sulfide was added slowly. Reconstituted proteins were desalted after 2 hr incubation on a PD-10 column equilibrated with reconstitution buffer. Incorporation of the Fe/S clusters into apoproteins was monitored by UV-Vis (V-550, Jasco Inc., Easton, MD) and CD spectroscopy (J-815, Jasco Inc).

## Miscellaneous procedures

Yeast mitochondria isolation was performed using the method of Glick and Pon (*Glick and Pon, 1995*). Standard procedures were performed for SDS-PAGE and immunoblotting. Anti-Sdh2 was from the previous study (*Kim et al., 2012*). BN-PAGE was performed as described previously with mitochondrial lysates in 1% digitonin solution (*Schilke et al., 1999*). Anti-Strep was purchased from Qiagen. Antibodies against LA-conjugated proteins were from Calbiochem (San Diego, CA). Anti-Myc and anti-HA were from Santa Cruz Biotechnology (Dallas, TX). Anti-Por1 was purchased from Molecular Probes and anti-FLAG was from Sigma-Aldrich.(St. Louis, MO) Protein concentration was determined by the Bradford assay.

# Acknowledgements

We thank James Cox and the University of Utah Metabolomics Core facility for the metabolomic analyses. We acknowledge support of funds in conjunction with grant P30 CA042014 awarded to Huntsman Cancer Institute. AM was supported by training grant T32 DK007115 from the National Institutes of Health. This research was supported by grants RO1 GM110755 and R01 GM112763 from the National Institutes of Health awarded to DRW and JAW, respectively. RL acknowledges generous financial support from Deutsche Forschungsgemeinschaft (SPP 1710 and SPP 1927), and the LOEWE program of state Hessen.

# Additional information

### Funding

| Funder | Grant reference number | Author |
| --- | --- | --- |
| National Institutes of Health | T32 DK007115 | Andrew Melber |
| Deutsche Forschungsgemeinschaft | SPP 1710 | Roland Lill |
| Deutsche Forschungsgemeinschaft | Spp 1927 | Roland Lill |
| National Institutes of Health | R01 GM112763 | James A Wohlschlegel |
| National Institutes of Health | RO1 GM110755 | Dennis R Winge |
| National Institutes of Health | P30 CA042014 | Dennis R Winge |

| LOEWE Zentrum für Synthe-
tische Mikrobiologie SynMikro | Roland Lill |

The funders had no role in study design, data collection and interpretation, or the decision to submit the work for publication.

### Author contributions

AM, Conception and design, Acquisition of data, Analysis and interpretation of data, Drafting or revising the article, Contributed unpublished essential data or reagents; UN, Acquisition of data, Analysis and interpretation of data, Drafting or revising the article; AV, BDW, Acquisition of data, Analysis and interpretation of data; RL, JAW, Analysis and interpretation of data; DRW, Conception and design, Analysis and interpretation of data, Drafting or revising the article

### Author ORCIDs

Roland Lill, http://orcid.org/0000-0002-8345-6518
Dennis R Winge, http://orcid.org/0000-0003-1160-1189

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
