## [Decision Letter]

Thank you for submitting your article "Role of Nfu1 and Bol3 in Iron-Sulfur Cluster Transfer to Mitochondrial Clients" for consideration by *eLife*. Your article has been favorably evaluated by Michael Marletta as the Senior editor and three reviewers, one of whom, Klaus Pfanner, is a member of our Board of Reviewing Editors. The reviewers have opted to remain anonymous.

The reviewers have discussed the reviews with one another and the Reviewing Editor has drafted this decision to help you prepare a revised submission.

Summary:

This paper by the Winge group is a rather comprehensive study of the role of Nfu1 and Bol1, 3 in the Fe-S cluster biosynthetic pathway confirming and extending the role of these proteins in the late delivery of 4Fe-4S to client enzymes. The study has been carefully done and quite clearly demonstrates especially the interaction of Nfu1 with the Isa complex and bol3 in Fe-S delivery. The role of Bol1 is less clear in this pathway as it is shown to interact with GRX5 and to bind 2Fe-2S clusters.

The manuscript is remarkably complete, although also remarkably complicated. It clearly establishes that Nfu1 and Bol3A in yeast function in a late stage in mitochondria 4Fe-4S protein maturation. The authors are careful to point out that it is not yet clearly established what the specific roles are and suggest that there could be more than one function. Of the two dominant possibilities (1) cluster transfer from the primary assembly complex to the target 4Fe-4S protein (2) serving as a stabilizer/redox buffer of the apo-target proteins, the authors appear to, reasonably, favor (1).

Given the complete and compelling nature of the data presented there is little to criticize from the technical perspective.

Essential revisions:

1) The Abstract contains many details, but is not easy to understand for readers outside the field. It contains several abbreviations that are not explained, e.g. ISA complex. The Abstract should be re-written and also include a last sentence that summarizes the main findings of the paper.

2) Although the data are presented clearly, the manuscript is difficult to read. The information is scattered in so many different figures, with much of the relevant data buried in the supplemental information, it is very difficult to keep track of the variously associated genotypes, phenotypes, and interacting partners. It seems to me that it would be very useful to the general reader if this work could be summarized in a table or two. In this way information in the table(s) could have superscripts that direct the reader to the relevant data presented in the various figures. The compelling aspect of the manuscript that justifies publication in *eLife* is that it provides clear biochemical linkage to known genetic defects in human patients, yet it will be good to structure the manuscript in a better digestible way.

3) The statement that Bol1 and Bol3 are redundant is not clear. Maybe I am missing something, but it seems that the double mutant only shows that they likely participate in the same pathway. If, as the authors intimate, Bol1 acts upstream of the Isa complex which generates 4Fe-4S clusters, then it might make sense that anything compromising this would have synthetic affects with downstream proteins, like Bol3 which are involved in delivering these clusters to client proteins.

4) I found the mass spectrometry data to be the most compelling. Nevertheless, this work is not, in my opinion, sufficiently emphasized. Also, it seems curious to me that the proteins implicated as partners are the only ones shown in the table. Unless there is something about the data presentation I missed, it seems highly unlikely to me that the expected interacting proteins are the *only* proteins that are consistently pulled down by this type of experiment. If that is true then the authors should state so explicitly, or they should have a more complete data set.

5) Using mass spectrometry analysis, interaction partners of Nfu1 could be detected. One of the proteins associated with Nfu1 was Lys4. This interaction was enhanced when the mutant Nfu1G/T>H (Nfu1m) was used as stated in Figure 5. However, in Figure 1 the pull-down using Nfu1m-Strep and decorated against Lys4-HA does not show a clear increase in association of Nfu1m with Lys4. The authors should comment on this.

---

## [Author Response]

*Essential revisions:*

*1) The Abstract contains many details, but is not easy to understand for readers outside the field. It contains several abbreviations that are not explained, e.g. ISA complex. The Abstract should be re-written and also include a last sentence that summarizes the main findings of the paper.*

The Abstract was completely rewritten.

*2) Although the data are presented clearly, the manuscript is difficult to read. The information is scattered in so many different figures, with much of the relevant data buried in the supplemental information, it is very difficult to keep track of the variously associated genotypes, phenotypes, and interacting partners. It seems to me that it would be very useful to the general reader if this work could be summarized in a table or two. In this way information in the table(s) could have superscripts that direct the reader to the relevant data presented in the various figures. The compelling aspect of the manuscript that justifies publication in eLife is that it provides clear biochemical linkage to known genetic defects in human patients, yet it will be good to structure the manuscript in a better digestible way.*

We considered many options to enhance the clarity of the figures and text. Besides the extensive rewriting of the text, we elected to modify the figure set to move certain supplemental figures to the main figure set. This will obviate the need for readers to move back and forth between the main paper and supplement. For Figure 3, we moved one key supplement to become panel Figure 3. This is a key figure to illustrate the effects of [4Fe-4S] defects on mitochondrial DNA translation via RNaseP. For Figure 4, we moved one supplement to the main figure as Figure 4. For Figure 5, we eliminated one supplemental figure that was redundant with the Figure 5 Supplemental tables. For Figure 6, we moved one supplement to the main figure to making inspection of the data easier. We generated a summary table as suggested, but found that too cumbersome. We are confident that moving these four supplemental panels to the main figures makes critical reading of the manuscript easier.

*3) The statement that Bol1 and Bol3 are redundant is not clear. Maybe I am missing something, but it seems that the double mutant only shows that they likely participate in the same pathway. If, as the authors intimate, Bol1 acts upstream of the Isa complex which generates 4Fe-4S clusters, then it might make sense that anything compromising this would have synthetic affects with downstream proteins, like Bol3 which are involved in delivering these clusters to client proteins.*

This is correct; they are not redundant proteins. The double mutant lacking both Bol1 and Bol3 exhibits a synthetic defect. We acknowledge that the text was not clear on this point. We revised the text in the Discussion to state the following:

"These studies suggest that Bol1 and Bol3 have specialized functions within the same pathway, such that cells lacking both Bol1 and Bol3 have a synthetic defect."

*4) I found the mass spectrometry data to be the most compelling. Nevertheless, this work is not, in my opinion, sufficiently emphasized. Also, it seems curious to me that the proteins implicated as partners are the only ones shown in the table. Unless there is something about the data presentation I missed, it seems highly unlikely to me that the expected interacting proteins are the only proteins that are consistently pulled down by this type of experiment. If that is true then the authors should state so explicitly, or they should have a more complete data set.*

We accumulated five MS data set with Nfu1 interactors. In each independent data set, there were many other proteins pulled down with WT Nfu1 or the Nfu1 mutant. A few proteins such as Pil1, Atp5 and Mdh1 were pulled down in two independent studies. However, in one of the studies Mdh1 was only pulled down by WT Nfu1 and not the enhanced interactor mutant Nfu1. Thus, the FeS client proteins were the only proteins pulled down reproducibly. With Bol1, the only proteins that were pulled down in all interaction studies were Grx5 and Prx1. The Prx1 peroxiredoxin was a reproducible hit that we did not include in the initial Table, since we wanted to focus an independent study on this interaction. However, we have done follow-up studies with Prx1 and cannot find a FeS defect in mutant cells. Thus, we have amended our Table to include Prx1. No other reproducible interactors are evident.

*5) Using mass spectrometry analysis, interaction partners of Nfu1 could be detected. One of the proteins associated with Nfu1 was Lys4. This interaction was enhanced when the mutant Nfu1G/T>H (Nfu1m) was used as stated in Figure 5. However, in Figure 1 the pull-down using Nfu1m-Strep and decorated against Lys4-HA does not show a clear increase in association of Nfu1m with Lys4. The authors should comment on this.*

The MS data was done in cells containing chromosomally expressed Lys4. In contrast, the study in Figure 1 was done in cells with Lys4-HA expressed from a high copy vector. In that situation, we do not see an enrichment of Lys4 with the Nful1 mutant likely due to a mass action effect. With normal Lys4 expression, the Nfu1 mutant exhibits enhanced interaction with Lys4 by MS.